

# Temporal variability and sources of VOCs in urban areas of Eastern Mediterranean

Christos Kaltsonoudis[1,2], Evangelia Kostenidou[1], Kalliopi Florou[1,2], Magda Psichoudaki[1,2], and Spyros N. Pandis[1,2,3]

[1]Institute of Chemical Engineering Sciences, ICE-HT, Patras, 26500, Greece
[2]Department of Chemical Engineering, University of Patras, Patras, 26504, Greece
[3]Department of Chemical Engineering, Carnegie Mellon University, Pittsburgh, 15213, USA

*Correspondence to*: Spyros N. Pandis (spyros@chemeng.upatras.gr)

**Abstract.** During the summer of 2012 volatile organic compounds (VOCs) were monitored by Proton Transfer Reaction – Mass Spectrometry (PTR-MS) in urban backgrounds sites, in Athens and Patras, two of the largest cities in Greece. Also during the winter of 2013 PTR-MS measurements were conducted in the center of the city of Athens. Positive matrix factorization (PMF) was applied to the VOC measurements to gain insights about their sources.

In summer most of the measured VOCs were due to biogenic and traffic emissions. Isoprene, monoterpenes and several oxygenated VOCs (oVOCs) originated mainly from vegetation either directly or as oxidation products. Isoprene average concentrations in Patras and Athens were 1 ppb and 0.7 ppb respectively while the monoterpene concentrations were 0.3 ppb and 0.9 ppb respectively. Traffic was the main source of aromatic compounds during summer. For Patras and Athens the average concentrations of benzene were 0.1 ppb and 0.2 ppb, of toluene 0.3 ppb and 0.8 ppb and of the xylenes 0.3 ppb and 0.7 ppb respectively.

Winter measurements in Athens revealed that biomass burning used for residential heating was a major VOC source contributing both biogenic compounds such as isoprene and the monoterpenes, and aromatic VOCs. Several episodes related to biomass burning were identified and emission ratios (ER) and emission factors (EF) were estimated.

## 1 Introduction

VOCs play an important role in atmospheric chemistry. Their reactions with hydroxyl (OH) radicals, ozone ($O_3$) and nitrate ($NO_3$) radicals produce secondary gas- and particulate-phase species (Atkinson, 2000; Hallquist et al., 2009). Many of the VOCs present in the atmosphere have adverse effects on human health (Kampa end Castanas, 2007) with exposure to compounds such as benzene, formaldehyde, and acetaldehyde having been related to cancer (Flesca et al., 1999). The European Commission has established an average annual limit of 1.5 ppb for benzene.





VOCs may originate from both natural and anthropogenic sources. Vegetation (Kesselmeier and Staudt, 1999; Goldstein and Galbally, 2007), volcanoes (Tassi et al., 2009; 2013) and marine emissions from the decay of organic matter (Kettle and Adreade, 2000; Meskhidze and Nenes, 2007; Colomb et al., 2008) are some of the natural sources that contribute significantly to the global VOC budget.

Anthropogenic emissions originate mainly from the use and production of fossil fuels, industrial processes and biofuel combustion. Biomass burning also contributes substantially both at the regional and global scales (Crutzen and Andreae, 1990; Karl et al., 2007; Koppmann, 2007).

On-line measurement techniques with high resolution, can provide valuable information about the sources of atmospheric pollutants (Goldstein and Schade, 2000; Millet et al., 2005; Slowik et al.,

2010). The PTR-MS, developed by Lindeger et al. (1998), can measure continuously the levels of a range of VOCs (DeGouw and Warneke, 2007).

Several campaigns focusing on VOC levels have been conducted in the Mediterranean basin. Holzinger et al. (2005b) reported secondary production of methanol and acetone in biomass burning plumes over the eastern Mediterranean during the MINOS campaign. PTR-MS measurements during summer in the

background site of Finokalia (Crete, Greece) (Salisbury et al., 2003) indicated methanol, acetone, acetonitrile, benzene and toluene concentrations of 3.3-6.1, 2.9-4.5, 0.15-0.44, 0.13-0.38 and 0.04-0.08 ppb respectively. Davison et al. (2009) studied the concentration and fluxes of biogenic VOCs in a Mediterranean ecosystem in west Italy during May-June 2007. Median concentrations of 1.6-3.5 ppb for methanol, 0.4-1.3 ppb for acetaldehyde, 1.0-2.0 ppb for acetone, 0.1-0.14 ppb for isoprene and 0.2-0.3

20   ppb for the monoterpenes were reported. Seco et al. (2011) compared the summer and winter VOC concentrations in a forest site 50 km away from Barcelona. Aromatic VOCs showed small variations between the 2 periods while biogenic VOCs (bVOCs) and oxygenated VOCs (oVOCs), were elevated during summer mainly due to higher physiological activity and faster photochemistry.

In addition real-time measurements have been reported for urban or semi-urban environments in

the Mediterranean. Filella and Penuelas (2006) studied the sources and variations of VOCs by PTR-MS at a semi urban site in Barcelona. Aromatic species such as toluene and benzene had diurnal patterns typical of traffic intensity. Their concentrations were higher during December and March. Biogenic species such as isoprene and monoterpenes had variable diurnal profiles during the four measurement periods. The isoprene diurnal cycle suggested considerable contribution from anthropogenic sources.

Monoterpenes had higher concentrations at night, decreasing in the morning until midday and increasing again in the evening. For some periods a peak also occurred during morning hours (06:00-09:00 LT), which was attributed to temperature-dependent emissions. Methanol, acetone and acetaldehyde levels were influenced by biogenic sources during summer. In a study at two sites in the city of Athens Rappengluck et al. (1998) monitored VOCs continuously over a 30-day late summer

period in 1994 by gas chromatography (GC). The $C_4$-$C_{12}$ hydrocarbons measured were strongly related to traffic emissions.





Source apportionment analysis has been applied to VOC data sets in order to quantify the contributions of the different VOC sources (Millet et al., 2005; Brown et al., 2007; Vlasenko et al., 2009; Yuan et al., 2012). Slowick et al. (2010) performed PMF analysis of a PTR-MS dataset as well as a unified AMS/PTR-MS dataset over a two week period in Toronto Canada. PTR-MS data included ten

mass to charge ratios (*m/z*). Factors related to traffic, long-range transport, local oxidation, and other sources were reported. Crippa et al. (2013) also performed PMF analysis on an AMS/PTR-MS unified dataset for the city of Paris during the MEGAPOLI project (summer 2009 and winter 2010 campaigns). The resulting common factors associated a large percentage of the aromatic VOCs with the hydrocarbon-like organic aerosol (HOA) during summer, while isoprene and the monoterpenes were

mainly related with the semi-volatile oxygenated organic aerosol (SV-OOA). For the winter period wood burning also contributed to the levels of the above species. Yuan et al. (2012) reported that reactions interfere with the PMF analysis of VOCs, proposing that one source can yield several factors at different stages of photochemical processing.

Despite the previous efforts, little is known concerning the current VOC sources in urban areas

of the eastern Mediterranean. Elevated PM concentrations have been detected in many urban areas (Pikridas et al., 2013) but the corresponding VOC emissions and concentrations have not been quantified. Enhanced sunlight intensity, higher temperature and $O_3$ concentrations during summer promote the emissions and oxidation of several VOCs. The aim of this work is to provide insights about the current VOC composition and origin in urban areas in the Eastern Mediterranean during summer

and winter. Another objective of the present study is to assess the impact of residential biomass burning for wintertime VOC levels in these urban areas.

## 2 Experimental

### 2.1 Sampling sites

Summer measurements in Patras (population 300,000) were conducted during 11 - 26 June 2012

in the Institute of Chemical Engineering Sciences (ICE-HT), located 8 km northeast of the city center (lat. 38.298, lon. 21.809, elevation 100 m). The area is surrounded by low vegetation and olive tree fields. The Athens (population 4 million) summer campaign was conducted between 3 and 26 July 2012 at the Demokritos National Center for Scientific Research in Athens (N.C.S.R.), which is located 8 km from the city center (lat. 37.995º, lon. 23.816º, elevation 280 m) at the foothills of a mountain covered

with pine vegetation. For the Athens winter campaign, the National Observatory of Athens (N.O.A.) (lat. 37.973, lon. 23.718, elevation 110 m) was selected due to its proximity to the city centre (<1.5 km). The selected urban background site is located on top of a small hill away from major city roads. The Athens winter campaign took place from January 9 to February 6, 2013. The locations of all sampling sites are presented in Figure 1.





### 2.2 Experimental setup

*Patras summer campaign*: VOC concentrations were monitored by a PTR-MS (PTR-QMS 500, Ionicon Analytik). A High Resolution–Time of Flight–Aerosol Mass Spectrometer (HR-ToF-AMS, Aerodyne Research) was used for the characterization of the aerosol composition. A Multi Angle Absorption Photometer (MAAP 5012, Thermo Scientific) monitored the black carbon (BC) concentration. A series of gas monitors was used for measuring nitrogen oxides ($NO_x$), $O_3$ and sulfur dioxide ($SO_2$) (API Teledyne, models T201, 400E and 100EU respectively). A Scanning Mobility Particle Sizer (SMPS) provided the number size distributions (TSI, models 3080 and 3787). Meteorological variables and sunlight intensity were also measured. The sampling was conducted at approximately 15 m above ground. All gas species were sampled through Teflon (PTFE) tubing while all particulate species were sampled through 3/8 in copper tubing.

*Athens summer campaign*: The same instrumentation (PTR-MS and HR-ToF-AMS) was used for the Athens summer campaign. BC concentration was provided by an aethalometer operating at 880 nm (Magee Scientific, AE31). An SMPS measured the particulate number distributions (model 3022 TSI, custom DMA). A $PM_{2.5}$ cyclone and a silica drier were installed prior to the SMPS and the aethalometer. The PTR-MS and AMS were measuring from inside the mobile laboratory of the Laboratory of Air Quality Studies (LAQS) parked next the N.C.S.R. station. Sampling for these instruments was conducted at 6 m above ground. For the PTR-MS sampling 1/4 in PTFE tubing was used. For the AMS 3/8 in copper tubing was selected. $O_3$ and $NO_x$ concentrations were measured by the Ministry of Environment, Energy and Climate Change at an adjacent (300 m away) station.

*Athens winter campaign*: The instrumentation was similar to that of the Patras summer campaign. Additionally carbon monoxide (CO) and carbon dioxide ($CO_2$) were monitored (API Teledyne, models 300E and T360 respectively). A Horiba monitor (model APSA 365) was used for the $SO_2$ measurements. Meteorological data were obtained by the National Observatory of Athens. All sampling inlets were approximately 6 m above ground.

*PTR-MS operation*: In all three campaigns, $H_3O^+$ was used as the reaction reagent for the PTR-MS. The drift tube was operated at a pressure of 2.2-2.3 mbar and its voltage was 600 V. The inlet flow was 0.5 L min$^{-1}$ and the inlet tube and reaction chamber were heated to 60 $^0$C. The corresponding ratio of the electric field strength to the gas number density was approximately 126 Td (1 Td = $10^{-17}$ V cm$^2$). Blanks were obtained at regular intervals through an activated carbon filter (Supelpure HC, Supelco). A teflon filter was installed before the sampling inlet. Concentrations were monitored in the mass scan mode with a total cycle time of 10 s. A calibration standard for the VOCs (Ionicon Analytik) was used. Concentrations are calculated based on:

$$[VOC] = CF_{VOC}\, C_{VOC} / C_{H3O+} \tag{1}$$

where, $CF_{VOC}$ is the calibration factor for each VOC and $C_{VOC}$ and $C_{H3O+}$ are the counts per second for the corresponding VOC and the primary ion respectively. The concentrations reported for the non





calibrated *m/z*'s are calculated for a reaction rate of K=2.0x10$^{-9}$ cm$^3$ s$^{-1}$. Table S1 in the Supplementary Material summarizes the VOCs measured during the three campaigns. Formaldehyde (*m/z* 31) was excluded from this dataset due to issues related to the humidity dependence of the measurements and its low proton affinity (deGouw and Warneke, 2007). Acetaldehyde (*m/z* 45) was also excluded from the
dataset due to negative values, probably resulting from $CO_2$ interferences (DeGouw and Warneke, 2007). Finally acrolein (*m/z* 57) and *m/z* 41 were excluded due to spikes associated to butanol emissions from the SMPS instruments. Details for the rest of the instrumentation used in the campaigns can be found elsewhere (Kostenidou et al., 2015; Florou et al., 2016).

**2.3 PMF analysis**

The PMF technique (Paatero and Tapper, 1994; Lanz et al., 2007) was used for the deconvolution of the PTR-MS data into factors. The analysis was performed using the PMF evaluation tool (Ulbrich et al., 2009) with Igor Pro 6.22A (Wavemetrics). The uncertainties were calculated based on Poisson ion counting statistics (de Gouw et al., 2003) and the measured background concentrations. The overall uncertainty is described by (DeGouw et al., 2003; Slowik et al., 2010):

$$\Delta(I - I_b) = \sqrt{\frac{I}{\tau} + \frac{I_b}{\tau_b}}$$  (2)

where, $I$ is the signal of the main measurements, $I_b$ is the signal from the background measurements, $\Delta(I-I_b)$ is the overall uncertainty, $\tau$ is the dwell time of the main sampling and $\tau_b$ is the dwell time of the background sampling. A total of 29 *m/z* values (all the *m/z* values in Table S1 but methanol) were used for the PMF analysis. Solutions with up to 10 factors were examined with an $f_{peak}$ ranging from -2.0 to
2.0 with a step of 0.2. The averaging time used was 5 minutes. The optimum solution for each measurement period was selected by evaluation of the model residuals, the mass spectra composition, and the correlations of the factor time-series with other measured pollutants.

**3 VOC concentrations and diurnal profiles**

**3.1 Patras summer 2012**

The overall measurement period can be divided into three sub-periods based on the prevailing meteorology, namely June 11 – 15, 16 – 22 and 23 – 26. During the first period SW winds prevailed with an average temperature of 24.4 $^o$C, a wind speed of 3 m s$^{-1}$ and 58% relative humidity. The second period was characterized by higher temperatures (28.4 $^o$C) and stronger (5.3 m s$^{-1}$) E-NE winds. The average relative humidity was 29%. The final period had SW winds mainly during the day and E-NE
winds during the night. The average temperature was 27.2 $^o$C and the average relative humidity was 49%. Average wind speed decreased during this period to 2.4 m s$^{-1}$. There was no precipitation during the campaign. Additional information about the meteorological conditions and solar radiation can be





found in the supplementary information (Figure S1). Back trajectory FLEXPART (Stohl et al., 2005) and Hysplit analysis (*Draxler and Rolph, 2013*) was performed (Kostenidou et al., 2015). The air masses were influenced mostly by the marine environment during the first period and by continental Greece during the second and third period.

Table 1 summarizes the average, median, upper and lower quartile of the concentrations for the calibrated VOCs along with other gas and particulate species. Methanol was the most abundant VOC with an average concentration of 10.2 ppb while acetone was the second most abundant with 2.9 ppb. Acetic acid had a mean concentration of 2.2 ppb. The *m/z* 69, reported here as isoprene had an average concentration of 1 ppb. Throughout the campaign, the acetonitrile levels were on average 0.1 ppb with a

flat diurnal profile, suggesting that biomass burning was not an important VOC source. Timeseries for the reported VOCs are presented in the SI (Figure S2). Maximum values for the aromatic compounds were in the range of 1 ppb for toluene and the xylenes while benzene concentrations up to 0.4 ppb were measured. Isoprene and the monoterpenes had peak concentrations up to 3 ppb and 1 ppb respectively at noon.

The average diurnal profiles for some of the VOCs measured are shown in Figure 2. The concentrations of isoprene, the monoterpenes, methyl vinyl ketone (MVK) and methacrolein (MACR) were low during nighttime and increased in the afternoon with a maximum at 15:00 LT– 16:00 LT. Other compounds of mainly biogenic origin such as methyl ethyl ketone (MEK) (Figure 2d) displayed a similar diurnal pattern, having two additional small peaks at 09:00 LT and 22:00, something that shows

either the influence of anthropogenic sources or the contribution of other molecules to the *m/z* 73 signal, reported here as MEK. Most of the biogenic VOCs had higher concentrations during the second and third period. During these periods the air masses reaching the site passed over continental Greece where there are mountains with forests.

Aromatic VOCs, like benzene, toluene and xylenes displayed diurnal cycles (Figures 2e, 2f, 2g)

characteristic of traffic emissions, with a peak during the morning rush hour at 08:00 LT and one peak in the evening at 22:00 LT. Higher concentrations were observed during the first period and during daytime for the third period (Figure S2). This is consistent with significant local aromatic sources. The toluene to benzene ratio for the overall period was 2.5. During the three periods, the toluene to benzene ratio was 2.3, 2.0 and 2.6 respectively. Most of the reported toluene to benzene ratios in the literature

are in the range of 1 or less (for aged air masses) to 4 (associated with fresh emissions measured in tunnels) (Roberts et al., 1984; Salisbury et al.; 2003, Kristensson et al., 2004). Since this ratio is considered to be a photochemical clock (Roberts et al., 1984) the sampled air during the second period was on average more aged. During this period the air masses sampled were less influenced by the city of Patras and the surrounding areas.

Methanol levels (Figure 2i) were affected by both anthropogenic and biogenic sources with higher values observed during the third period. Methanol's diurnal cycle had peaks during morning and




evening rush hours. Acetone's concentrations were marginally elevated during daytime from 13:00 LT–22:00 LT (Figure 2j). Several oVOCs had a diurnal pattern similar to the biogenic VOCs. The diurnal profiles of formic acid/ethanol (Figure 2k) and acetic acid (Figure 2l) peaked at approximately 17:00 LT. Higher values of the oVOCs were observed during the second period (SI-Section 2, Figure S2) suggesting strong biogenic influence.

NO$_x$ and BC concentrations displayed diurnal profiles (Figures S3 and S4) similar to those of the aromatic species, typical of traffic emissions. Higher values were observed for both species during the first and third periods, while for the second period lower values were measured due to the strong E-NE winds. The average concentration for NO$_x$ was 5.1 ppb, while for BC a mean value of 0.5 µg m$^{-3}$ was measured. SO$_2$ concentrations were elevated during the day. A peak during morning rush hour, similar to the aromatic VOCs, was observed but no peak was evident during evening rush hour. This can be explained by the low activity of heavy trucks during these hours (22:00-23:00 LT). A mean value of 1.3 ppb was found for SO$_2$. Ozone concentrations were elevated during the second period exceeding 60 ppb (8 h average) during the night of the 17 to 18 of July. These elevated O$_3$ levels were the result of long range transport from continental Greece and the Balkans. During these two days an average value of 55 ppb was measured. Diurnally averaged O$_3$ concentrations for the overall measurement period started to increase at 9:00 LT with a maximum of 58 ppb during 15:00-16:00 LT. During the night (21:00-7:00 LT) O$_3$ concentrations were in the range of 45 ppb.

**3.2 Athens summer 2012**

The Athens summer campaign was characterized by a prolonged heat wave with temperatures reaching up to 40 °C. The average temperature was 29.6 °C and the corresponding relative humidity was 40%. Local topography dictated the wind patterns, with the wind having mainly a E-SE direction. The wind direction varied especially during the morning hours (08:00-11:00 LT) with wind shifts bringing air masses from the N or E of the site. The average wind speed was 1.7 m s$^{-1}$ with very low values during the night. No precipitation occurred throughout the campaign. More information concerning the meteorological conditions is included in the SI (Figure S5).

Methanol was the most abundant VOC with an average value of 6.7 ppb (Table 2). The second most abundant was acetone with a mean concentration of 4.3 ppb, significantly higher than in Patras. Throughout the campaign the acetonitrile concentrations were on average 0.2 ppb with a flat diurnal profile, suggesting the lack of biomass burning sources. Figure S6 includes all the VOC time series for the Athens summer campaign. Toluene and the xylenes peaked during the morning hours at concentrations in the range of 3 ppb, while benzene had typical maximum values around 1 ppb during these hours. Isoprene and the monoterpenes peaked during noon at concentrations in the range of 1 and 2 ppb respectively.



The average isoprene concentration was 0.7 ppb with a diurnal profile (Figure 3a) similar to Patras. Local sources influenced significantly the monoterpene concentrations (Figure 3b) with a peak during the morning hours (from 06:30 to 08:30) followed by several hours of elevated levels with a maximum at approximately 15:00-16:00 LT. This behavior is similar to that reported by Filella and

5 Penuelas (2006) for the Barcelona semi-urban site. Nighttime stagnation conditions followed by morning N or NE winds caused these spikes by transporting to the site nighttime emissions from the adjacent pine forest. MVK and MACR (Figure 3c) had a similar diurnal cycle to that of isoprene, while MEK (Figure 3d) peaked at 11:00-12:00 LT and had one more peak at 22:00, something seen also in the Patras summer campaign.

In Athens, higher aromatic concentrations were observed compared to the Patras summer campaign.. These species had similar diurnal cycles (Figures 3e, 3f, 3g) characteristic of traffic emissions, with a peak during the morning rush hour at 8:00-9:00 LT and one wide peak during the evening from 19:30 to 1:00 LT. The toluene to benzene ratio for the Athens summer campaign was 3.4, a value similar to that of fresh traffic emissions (Kristensson et al., 2004). Rappenglueck et al. (1998)

reported a toluene to benzene ratio of 2.3 for the Demokritos site during August- September 1994. The correlation coefficients between the aromatic species (benzene, toluene, xylenes, C9 and C10 aromatics) were high ($R^2$ ranging from 0.92 to 0.96) suggesting a single source related to traffic emissions.

The diurnal profiles of most oVOCs, suggest anthropogenic influence (Figures 3i, 3j, 3k and 3l). Methanol had a similar pattern to that of the aromatic species with peaks at 08:00 LT and 22:00 LT.

Acetone concentrations were elevated during the day with additional peaks during morning rush hour and during night (24:00 LT). Formic acid/ethanol peaked during the afternoon hours, at 15:00-16:00 LT, similar to the biogenic VOCs. Acetic acid concentrations were lower during the day with a peak at 22:00-23:00 LT and elevated concentrations during night hours. Morning rush traffic hour had a very small effect on acetic acid's levels. This compound appears to be associated mainly with regional

sources.

$NO_x$ and BC concentrations displayed diurnal profiles similar to those of the aromatic species, typical of traffic emissions (Figures S7 and S8). The average concentration of $NO_x$ was 7.3 ppb, while BC had an average value of 0.7 $\mu g\ m^{-3}$. Increased solar radiation along with the availability of $O_3$ precursors resulted in high $O_3$ exceeding the EU 60 ppb 8-hour limit in 4 days during the 13 day period.

The $O_3$ average diurnal pattern was similar to that of Patras during the summer with a maximum of 60 ppb during 15:00-16:00 LT. During the night concentrations of 50 ppb were measured.

**3.3 Athens winter 2013**

Due to the recent economic crisis and the increasing prices of heating oil at the time, a significant percentage of the population in Greece has been using wood for residential heating during

the winter. During the winter of 2012-2013, extreme episodes of atmospheric pollution occurred in





Athens, with hourly $PM_1$ levels reaching values up to 140 µg m$^{-3}$ (Florou et al., 2016). The average temperature, humidity, and wind speed during the Athens winter campaign were 11.9 $^o$C, 70%, and 3.4 m s$^{-1}$ respectively. The wind direction patterns favored mainly S-SW winds and occasionally N-NE winds. Minimum night temperatures ranged from 3.3 $^o$C to 15 $^o$C. Precipitation occurred during January 11, 16, 18, 24, 25 and 26. Meteorological data are included in the SI (Figures S9 and S10).

Table 3 summarizes the average, median, lower and upper quartiles of the concentrations for the measured VOCs. The substantial difference between the median and average values for most species is due to periods with significant residential wood burning and corresponding high concentrations. Extreme concentrations were measured with methanol reaching up to 90 ppb, acetone and acetic acid reaching 15 ppb, and isoprene reaching 9 ppb were measured. Methanol, acetone, acetic acid and toluene had concentrations with median values ranging from 2.4 to 1.3 ppb. Methanol and acetone levels were significantly lower compared to the summer campaign. Formic acid/ethanol and acetic acid had similar concentrations with the summer. The timeseries for the measured VOCs is shown in Figure S11 in the SI.

Most of the VOCs (Figure 4) had higher average concentrations during the night and during the morning rush hour (peak at 09:00 LT). Evening concentrations started to increase at 19:00 LT. Very high concentration periods were detected for most of the measured VOCs. Aromatics increased during the morning rush hour together with formic acid and acetone. The toluene to benzene ratio for the campaign was 2.1 suggesting the importance of other sources in contrast to the summer ratio (3.4) when traffic dominated.

Isoprene and the monoterpenes had elevated concentrations during the morning but also during the evening (18:00-23:00 LT) (Figures 4a and 4b) mainly due to biomass burning. MVK, MACR and MEK were also produced by biomass burning sources (Figures 4c and 4 d). Although these compounds dominate their specific $m/z$ values ($m/z$: 69, 71, 73) in typical rural environments, other compounds may be present in these $m/z$ values during the Athens winter campaign. Karl et al. (2007) reported furan and other alkenes being detected at $m/z$ 69, crotonaldehyde and other alkenes for $m/z$ 71 and methyl propanal for $m/z$ 73 during the burning of tropical biofuels.

$NO_x$, BC and CO had similar timeseries and diurnal profiles (Figures S12 and S13) as the majority of the VOCs with peaks at morning rush hour and during the evening (18:00-23:00 LT). Their median values were 11.3 ppb, 1.0 µg m$^{-3}$ and 100 ppb respectively. $SO_2$ concentrations were elevated mainly during the morning rush hour. During night lower $SO_2$ values were observed with some peaks around 23:00 LT that can be attributed to traffic and residential heating using diesel fuel. A median value of 1.1 ppb was observed. $O_3$ levels were reduced during the biomass burning periods to almost zero. For periods with stronger winds and low gas and particulate pollutant levels, the $O_3$ concentrations reached 30 to 40 ppb. The average $O_3$ diurnal pattern had a peak at 15:00 LT and a median value of 25 ppb for the campaign.




Acenonitrile (*m/z* 42) and levoglucosan's tracer (*m/z* 60 of the AMS) were used as markers in order to identify periods where wood burning had a substantial contribution to the VOC and particulate levels. The concentration of these two species (Section S4.1) had a relatively high correlation ($R^2$=0.78). In both cases elevated concentrations were observed mainly during the night, but also during

the morning having a peak at 9:00 LT. These morning peaks can be attributed to residential heating at these hours but also can be a result of vertical mixing of the atmosphere. The acetonitrile's relative increase is higher than of the levoglucosan tracer (Figure 5) during the morning rush hour. Recently Dunne et al. (2012) reported interferences at PTR-MS *m/z* 42 under the $H_3O^+$ reaction when measuring in polluted urban environments. These interferences were attributed to the $^{13}C$ isotopologues of $C_3H_5^+$

and the product ion $C_3H_6^+$ formed by reactions with $NO^+$ and $O_2^+$ that exist in trace amounts in the reagent gas ($H_3O^+$). This leads to the conclusion that even though an increase due to biomass burning is apparent during the morning hours (increase in levoglucosan's tracer from the AMS), a fraction of *m/z* 42 concentration is due to species other than acetonitrile. During the morning, traffic appears to contribute significantly to the *m/z* 42 increase.

A set of criteria was established in order to select periods during which residential heating dominated the gas and particulate composition of ambient air. These periods (Table S2) were during nighttime (18:00-06:00 LT) and had acetonitrile concentrations exceeding 0.25 ppb. During these periods NW winds dominated bringing air masses from the northern parts of Athens to the site (Figure S10b). The average concentrations for these periods are shown in Table 3.

Excluding the above biomass burning periods from the campaign dataset resulted in a decrease of the average values of the measured species by 11-34%. Namely the average values decreased by 34% for methanol, 26% for acetonitrile, 11% for formic acid and ethanol, 17% for acetone, 16% for acetic acid, 23% for isoprene, 25% for MVK and MACR, 17% for MEK, 23% for benzene, 18% for toluene, 22% for the xylenes and 24% for the monoterpenes.

**4 Biomass burning emissions**

Emission ratios for the gas and particulate species were calculated as Δ[Species]/ΔCO (Table 4) where Δ[Species] is the difference in the concentration, of each compound, between the burning period and an adjacent reference period. This reference period was selected based on the lower acetonitrile, CO, and $CO_2$ concentrations at adjacent time periods (prior or later) to the period identified as biomass

burning. The estimated ratios are given in Table 4. Emission ratios for toluene, xylenes, C9 and C10 aromatics seem to be greater than various SE, SW and pine spruce fuels (Warneke et al., 2011). Acetic acid's and acetonitriles's emission ratios were lower than the values reported by Warneke et al. (2011). The modified combustion efficiency (MCE) calculated for these biomass burning periods was high (0.975 – 0.983) indicating low contribution from smoldering phase burning.

Emission factors (EF) were calculated based on:



$$EF_i = \frac{(C_i)_{plume} - (C_i)_{background}}{(CO_2)_{plume} - (CO_2)_{background}} EF_{CO_2} \qquad (3)$$

where $(C_i)_{plume}$ and $(C_i)_{background}$ were the concentrations for species i, during the burning and the reference periods respectively and $(CO_2)_{plume}$ and $(CO_2)_{background}$ are the concentrations of $CO_2$ during the biomass burning and the reference periods. For the emission factor of $EF_{CO_2}$ a value of 1600 g kg$^{-1}$ was used. This value was selected based on the typical values reported in the literature for several fuel types. The estimated EFs for the measured species are shown in Table 4.

## 5 VOC sources

### 5.1 Patras summer 2012 - PMF

For the Patras PMF analysis of the PTR-MS data a four factor solution was selected. This choice is further discussed in the SI (Sections 5.1 and 5.2). An $f_{peak}$ of zero was chosen based on the $m/z$ composition of the factors and the correlations of the factors to known external (BC and $NO_x$) and internal (aromatic VOCs, biogenic VOCs) tracers (SI - Section 5.3). Solutions in the $f_{peak}$ range from -0.4 to 0.2 have only minor differences. Other solutions deriving from different $f_{peak}$ selections are also presented in the SI. The factors obtained were attributed to biogenic emissions (factor bVOC), traffic emissions (factor TRAF), oxygenated VOCs (factor OVOC) and oxygenated VOCs of biogenic origin (factor b-OVOC).

The bVOC factor included mainly isoprene ($m/z$ 69) and the monoterpenes ($m/z$ 137) (Figure 6). There were small contributions by $m/z$ values 71, 81, 87 and 101. The factor average diurnal pattern has a peak at 15:00 LT. The factor showed weak correlations with the biogenic oxygenated organic aerosol (b-OOA) and the moderately oxygenated OA (M-OOA) ($R^2$=0.20 and 0.22 respectively) as obtained by the PMF analysis of the AMS data (Kostenidou et al., 2015). This weak correlation is due the fact that the b-OOA has been at least partially produced away from the measurement site. This highlights the challenges of combining gas and particulate phase measurements during source apportionment applications. Sunlight intensity had also a weak correlation ($R^2$ = 0.24) with the bVOC factor. The factor also showed weak correlation ($R^2$ = 0.29) with ambient temperature. Shifting the time series of the factor earlier by two hours, marginally improved this correlation ($R^2$ = 0.35) with temperature and with sunlight intensity ($R^2$ = 0.28). This suggests that some of the biogenic components included in this factor are transported emissions to the site from nearby areas.

Factor TRAF included the aromatic species ($m/z$ values 79, 93, 107, 121 and 135) as well as $m/z$ values 43, 59, 61 and 69 (Figure 6). Some contribution by $m/z$ values 71 and 73 was also observed. Its diurnal profile as expected is similar to that of the aromatic species (Figure 2) with one peak at 09:00 LT and one at 22:00 LT. The toluene to benzene ratio of the factor was 2.6 the same as the ratio for the



third period. Correlations between this factor and BC were relatively high ($R^2$ = 0.66). The two hydrocarbon-like OA factors (HOA-1 and HOA-2) reported by Kostenidou et al. (2015) showed $R^2$ of 0.49 and 0.43 with this factor respectively. The PM$_1$ organonitrate calculated for the AMS measurements based on Farmer et al. (2010) also correlates well ($R^2$ = 0.59) with the TRAF factor. The

main source of this factor based on the diurnal profiles and the correlations seen is traffic in the city of Patras and the surrounding areas. Other sources such as cooking emissions, ships, etc., might contribute to this factor to some extent.

Factor OVOC mainly included acetone. Other *m/z* values contributing to this factor were 43, 47 and 61 (Figure 6). A weak correlation between the factor and the AMS's b-OOA ($R^2$=0.29) was

observed. Additionally low correlations were observed with particulate sulfate and nitrate ($R^2$ of 0.23 and 0.18 respectively). The factor's diurnal pattern (Figure 6) reveals elevated concentrations during the night and a minimum at 15:00 LT. The factor seems to be affected by mixing height changes and from long range transport.

The last factor, (b-OVOC) included oxygenated molecules (acetic acid, acetone and formic

acid/ethanol). Other *m/z* values contributing to this factor were *m/z* 43, 69, 71, and 73 (Figure 6). The b-OVOC diurnal cycle is similar to that of the bVOC factor shifted by approximately 2 hours. Based on this diurnal cycle, these b-OVOC species can be considered as either products of the oxidation of biogenic molecules or direct biogenic emissions. Ambient temperature and sunlight intensity data had a mild correlation with the b-OVOC with $R^2$ of 0.35 and 0.32 respectively. A weak correlation ($R^2$ = 0.27)

of this factor was seen with the moderately oxygenated (M-OOA) by the AMS and no correlation ($R^2$ = 0.08) was observed with the b-OOA. This supports the explanation that the b-OVOC factor is mostly oxidation products of local biogenic VOCs as well as temperature sensitive emissions.

The contribution of each factor to the various VOC species is presented in Figure S21. 81% of isoprene (*m/z* 69) is included in the biogenic factors bVOC and b-OVOC. Monoterpene *m/z* values (137

and 81) were mainly assigned by the PMF to the bVOC factor (73% and 63% respectively) and at smaller percentages to factor b-OVOC (9% and 6% respectively). Factor TRAF included the majority of the aromatic species (73 – 91 %). The rest of the aromatic species are assigned by the PMF to other factors. This is probably due to the combination of the low concentrations of most of the aromatics and the uncertainties of the PMF analysis. These values also provide an estimate of the uncertainty of these

results. Patras summer PMF analysis demonstrates the importance of biogenic sources to the VOC budget of urban areas during summer.

## 5.2 Athens summer 2012 - PMF

After the evaluation of various solutions, five factors were selected for this period with an *f*$_{peak}$ equal to zero (SI - Sections 6.1, 6.2, 6.3). The factors were attributed to monoterpene species (factor



TERP), other biogenic emissions (factor bVOC), traffic emissions (factor TRAF) and two oxygenated VOC factors (factor OVOC-1 and OVOC-2).

For the Athens summer campaign, the local monoterpene emissions were separated by the PMF from the rest of the biogenic emissions creating a new factor named TERP. The $m/z$ values 137 and 81 were the main components of this factor and its diurnal profile (Figure 7) has an early morning peak similar to that of the monoterpenes in this site (Figure 3). Other $m/z$ values contributing to this factor were 43, 47, 59, 61, 69, 71, 73, 75, 77, and 93 (Figure 7). No correlations ($R^2 < 0.05$) were found between this factor and measured particulate species. These monoterpene emissions were due to local pine vegetation and their concentrations were affected by local mixing phenomena.

Factor bVOC was mainly composed of isoprene ($m/z$ 69) and isoprene hyperoxides ($m/z$ 101). Also a fraction of $m/z$ values 43, 47, 59, 71, 73, 75, 87, 93, and 121 was included. Its diurnal profile (Figure 7) peaked at 15:00 LT, similar to the bVOC factor for the Patras summer campaign. This factor had a weak correlation ($R^2=0.15$) with the very oxygenated OA factor (V-OOA) obtained by the PMF analysis of the AMS data (Kostenidou et al., 2015). No correlation existed with the AMS M-OOA factor.

Factor TRAF included the aromatic species ($m/z$ values 79, 93, 107, 121 and 135). Also $m/z$ values 43, 47, 59, 61 and 73 contributed to this factor (Figure 7). The diurnal profile is similar to that of the aromatic compounds (Figure 3) with one peak at 09:00 LT and one at 22:00 LT. The toluene to benzene ratio of the factor was 3.1 indicating relatively fresh traffic emissions. Factor TRAF had moderate to high correlations with most of the species emitted by transportation sources: $NO_x$ ($R^2=0.69$), BC ($R^2=0.45$), AMS nitrate ($R^2=0.64$) and AMS HOA-2 ($R^2=0.58$). The factor's correlation with AMS HOA-1 was surprisingly weak ($R^2=0.16$). This inconsistency is believed to derive from the inhomogeneity of the source distribution around the sampling site along with the shifting wind directions (Kostenidou et al., 2015). Particulate organonitrates (AMS) showed good correlation with factor TRAF ($R^2=0.65$).

Factor OVOC-1 was dominated by acetone. Other $m/z$ values contributing to this factor were 43, 47 and 61 (Figure 7). No correlation ($R^2 < 0.08$) was found between this factor and other species. Its concentration had a modest afternoon peak. This factor is considered a mix of contributions by long range transport and various anthropogenic and biogenic sources.

The last factor (OVOC-2) included mainly acetone ($m/z$ 59) and acetic acid ($m/z$ 61). The $m/z$ 43 and 47 also contributed (Figure 7). The factors concentrations were elevated during the night. The factor had correlations with $PM_1$ nitrate ($R^2=0.49$), HOA-1, and HOA-2 ($R^2$ equal to 0.18 and 0.22 respectively) and also with V-OOA ($R^2=0.34$) and $PM_1$ sulfate ($R^2=0.32$). Due to these correlations and its diurnal profile, this factor is believed to be linked to a mix of urban sources as well as long range transport.

The fraction of some selected VOC species attributed to each factor is shown in Figure S31 (SI). The majority (58%) of isoprene is assigned to factor bVOC and smaller percentages to factors OVOC-1,




TERP and TRAF (14, 20 and 7% respectively). Monoterpenes, $m/z$ values 137 and 81, are almost exclusively included in factor TERP (84 and 76% respectively). The majority (55 to 73 %) of the aromatic species ($m/z$ values 79, 93, 121 and 135) are part of factor TRAF.

### 5.3 Athens winter 2013 - PMF

A five factor solution and an $f_{peak}$ equal to zero were selected for the Athens winter campaign (SI sections 7.1, 7.2 and 7.4). One factor was related to emissions originating from biomass burning (BBVOC). A factor for traffic emissions (factor TRAF) was identified and also one factor that was attributed mainly to industrial sources (factor IND). Finally two oxygenated VOC factors (OVOC-1 and OVOC-2) were found. The measurement period that was selected for this PMF analysis did not include

the first day of the measurements of the Athens winter campaign. Including the first day in the Athens winter campaign in the PMF analysis, resulted in the separation of the biomass burning source into two or more factors for solutions with 4 or more number of factors (SI Section 7.3). No distinct variation was seen for the favored wind direction between those BBVOC. Their timeseries (Figure S42) though similar showed significant differences for the first day of the campaign (Jan. 9 and Jan. 10) where lower

temperatures were recorded.

     Factor BBVOC correlated strongly with the AMS $m/z$ 60 ($R^2$=0.92) linked to levoglucosan and also with acetonitrile ($R^2$=0.80). It also had a high correlation ($R^2$=0.84) with the AMS BBOA factor (Florou et al., 2016) and with other species emitted by biomass burning sources such as CO ($R^2$=0.59) and BC ($R^2$=0.63). The factor mainly included acetone, isoprene, MVK and MACR, MEK, benzene,

xylenes, C9 aromatics and monoterpenes (Figure 8). Due to the acetonitrile presence along with the high correlations obtained with known species of biomass burning origin (BBOA, AMS $m/z$60, CO, BC), this factor was considered to derive from emissions of biomass burning. The diurnal profile (Figure 8) of the factor is characteristic of residential heating use: mild use during the early morning hours and high use during the night (from 18:00-24:00 LT).

Factor TRAF has the same $m/z$ values as in the summer campaigns and is mainly linked to traffic. The aromatics are the main VOCs contributing to this factor. Other $m/z$ values that are present in considerable amounts are: 43 and 59 (Figure 8). The toluene to benzene ratio for this factor is 3.0, similar to the value (3.1) of the same factor during the summer campaign in Athens. Its diurnal profile includes one large peak corresponding to the morning rush hour and a smaller peak at approximately

19:00 LT (Figure 8). The three factors related to traffic (factors named TRAF for the three studied campaigns) had a quite consistent $m/z$ spectrum with $R^2$ values ranging from 0.68 to 0.90.

Factor IND mainly includes toluene and acetone (Figure 8). Its diurnal pattern is relatively flat with slightly elevated concentrations during the night. A weak minimum is observed at 09:00 LT. Due to the presence of toluene this factor is considered of anthropogenic origin, probably related to industrial

activities far from the site.





Factor OVOC-1 includes mainly formic acid (*m/z* 47) and some isoprene. The factor's diurnal profile (Figure 8) was characterized by higher values during the early morning hours, peaking at 8:00 LT. After a significant decline in the morning it slowly increased during the rest of the day. No significant correlations were found between this factor and other measured species.

The last factor (OVOC-2) includes *m/z* 43 and acetic acid. The diurnal profile had elevated values during the night hours and was slowly decreasing through the day. It is believed to be associated with background concentrations of several VOCs (Figure 8).

For this measuring period, over 35% of benzene, 14% of the xylenes, 13% of the C9 aromatics and 14% of the C10 aromatics is estimated to be due to residential biomass burning. Additionally traditionally biogenic species such as isoprene and the monoterpenes were originating mainly (46 and 36% respectively) from residential biomass burning during this winter period. The Athens winter PMF analysis highlights the importance of biomass burning during winter for the VOC budget. Biomass burning and traffic were the major sources for the majority of the VOCs that were measured. On the contrary biogenic sources had a limited contribution to the VOC budget.

## 6 Conclusions

On-line measurements of VOCs and other gas and particulate species were performed in background urban sites of Greece during the summer and winter of 2012 and 2013. For the summer campaigns in Patras and Athens, the isoprene concentration was on average 1.0 and 0.7 ppb respectively. The corresponding values for the monoterpenes were 0.3 and 0.9 ppb with the elevated Athens concentrations originating mainly from local emissions. Typically the isoprene concentrations peaked during the day with values up to 2 ppb, while during the night the concentrations were lower (<0.5 ppb). Benzene, toluene and the xylenes average concentrations were 0.12, 0.28 and 0.25 ppb for Patras and 0.22, 0.81 and 0.67 ppb for Athens respectively. These differences are considered modest if the population difference (one order of magnitude) is taken into account. During the morning rush hour aromatic compounds such as toluene, peaked up to 5 ppb for Athens during the summer and 1 ppb for Patras. Analogous concentrations were seen for the xylenes, while benzene values during rush hour were in the range of 1 ppb for Athens and 0.3 ppb for Patras.

During winter in Athens the average concentrations for isoprene and the monoterpenes were similar to the summer values (1.1 ppb for isoprene and 0.4 ppb for the monoterpenes) indicating the importance of biomass burning sources of these compounds. Benzene, toluene and the xylenes concentrations were elevated during winter (1.0 ppb, 2.3 ppb and 1.7 ppb respectively) compared to summer, something attributed to the lower mixing height and the presence of additional sources. Concentrations up to 90 ppb for methanol and up to 15 ppb for acetic acid, toluene and the xylenes were measured. For the majority of the measured species elevated concentrations were observed during night signifying the importance of biomass burning, which was enhanced by the lower mixing heights.



PMF analysis showed that traffic was the major aromatic VOC source in all three campaigns. A traffic related factor was identified including more than 60% of benzene, 60% of toluene, 70% of the xylenes and 70% of the C9 aromatics. Especially during summer traffic dominated the above aromatic budget by contributing 73% of benzene for Patras and Athens, and 80% and 62% of toluene for Patras and Athens respectively. 85% of the xylenes in Patras and 70% in Athens were also apportioned to traffic.

The recent economic crisis along with the higher prices of fossil fuels has led to increased use of biomass burning for residential heating. Biomass burning emission ratios (ER) and emission factors (EF) due to residential heating were estimated. PMF analysis showed that the emissions of the aromatics by biomass burning were comparable to traffic emissions. Approximately 35% of benzene was due to biomass burning during the Athens winter campaign. Additionally during winter, the 'traditional' biogenic species, such as isoprene, the monoterpenes, MVK and MACR were originating mainly from biomass burning processes. During summer these biogenic species were almost exclusively linked to emissions from vegetation. During summer several oVOCs such as acetic acid, formic acid and ethanol were also linked to biogenic emissions either directly or as secondary products.

## Acknowledgements

We thank Prof. Argyriou (Department of Physics - University of Patras) for providing the meteorological data, Dr. Eleutheriadis (N.C.S.R. "Demokritos") for providing the facilities for the Athens summer campaign, Dr. Gerasopoulos (National Observatory of Athens - N.O.A) for providing the facilities for the Athens winter campaign. This research was co-financed by Greece and the European Union through the education and lifelong learning operational program Thales: "Sources and physicochemical properties of fine and ultrafine aerosol particles that affect the regional climate of Greece". This research was also supported by the European Research Council Project ATMOPACS (Atmospheric Organic Particulate Matter, Air Quality and Climate studies) (Grant Agreement 267099) in coordination with project PEGASOS.

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



**Table 1:** Concentrations (5 minute averages) of VOCs and other pollutants during the Patras summer campaign.

| | | Lower quartile | Median | Average | Upper quartile |
|---|---|---|---|---|---|
| **VOCs** | *m/z* | Concentration (ppb) | | | |
| Methanol | 33 | 7.01 | 8.34 | 10.18 | 10.67 |
| Acetonitrile | 42 | 0.08 | 0.10 | 0.12 | 0.14 |
| Formic acid, Ethanol | 47 | 0.40 | 0.86 | 0.89 | 1.35 |
| Acetone | 59 | 2.05 | 2.44 | 2.93 | 3.23 |
| Acetic acid | 61 | 1.16 | 1.88 | 2.15 | 2.90 |
| Isoprene | 69 | 0.32 | 0.85 | 1.01 | 1.46 |
| MVK+MACR | 71 | 0.16 | 0.24 | 0.30 | 0.38 |
| MEK | 73 | 0.18 | 0.25 | 0.30 | 0.42 |
| Benzene | 79 | 0.06 | 0.09 | 0.12 | 0.17 |
| Toluene | 93 | 0.10 | 0.19 | 0.28 | 0.43 |
| Xylenes | 107 | 0.09 | 0.15 | 0.25 | 0.37 |
| Monoterpenes | 137 | 0.09 | 0.19 | 0.33 | 0.50 |
| **Other species** | **Units** | | | | |
| Black carbon (BC) | μg m$^{-3}$ | 0.27 | 0.44 | 0.49 | 0.67 |
| Sulfur dioxide (SO$_2$) | ppb | 0.98 | 1.15 | 1.25 | 1.41 |
| Nitrogen oxides (NO$_x$) | ppb | 1.8 | 3.7 | 5.1 | 6.4 |
| Ozone (O$_3$) | ppb | 41.2 | 49.2 | 48.7 | 56.4 |





**Table 2:** Concentrations (5 minute averages) of VOCs and other pollutants during the Athens summer campaign.

| | | Lower quartile | Median | Average | Upper quartile |
|---|---|---|---|---|---|
| **VOCs** | *m/z* | Concentration (ppb) | | | |
| Methanol | 33 | 5.26 | 6.13 | 6.72 | 8.13 |
| Acetonitrile | 42 | 0.16 | 0.19 | 0.20 | 0.23 |
| Formic acid, Ethanol | 47 | 1.12 | 1.51 | 1.52 | 1.87 |
| Acetone | 59 | 3.27 | 4.09 | 4.28 | 4.92 |
| Acetic acid | 61 | 1.53 | 2.04 | 2.17 | 2.65 |
| Isoprene | 69 | 0.46 | 0.72 | 0.73 | 0.93 |
| MVK+MACR | 71 | 0.20 | 0.32 | 0.35 | 0.47 |
| MEK | 73 | 0.32 | 0.45 | 0.50 | 0.59 |
| Benzene | 79 | 0.12 | 0.16 | 0.22 | 0.24 |
| Toluene | 93 | 0.48 | 0.62 | 0.81 | 0.82 |
| Xylenes | 107 | 0.35 | 0.51 | 0.67 | 0.77 |
| Monoterpenes | 137 | 0.54 | 0.88 | 0.92 | 1.22 |
| **Other species** | **Units** | | | | |
| Black carbon (BC) | $\mu g\ m^{-3}$ | 0.43 | 0.60 | 0.72 | 0.89 |
| Nitrogen oxides ($NO_x$) | ppb | 5.1 | 5.6 | 7.3 | 7.5 |
| Ozone ($O_3$) | ppb | 49.6 | 55.5 | 54.1 | 59.1 |





**Table 3:** Concentrations (5 minute averages) of VOCs and other pollutants during the Athens winter campaign. Values are shown for the overall period and for the biomass burning periods.

| | | Overall measurement period | | | | Biomass burning periods | | | |
|---|---|---|---|---|---|---|---|---|---|
| | | Lower quartile | Median | Average | Upper quartile | Lower quartile | Median | Average | Upper quartile |
| **VOCs** | *m/z* | Concentration (ppb) | | | | | | | |
| Methanol | 33 | 1.23 | 2.42 | 6.07 | 5.91 | 13.78 | 19.68 | 25.65 | 30.51 |
| Acetonitrile | 42 | 0.05 | 0.08 | 0.16 | 0.18 | 0.34 | 0.45 | 0.55 | 2.70 |
| Formic acid, Ethanol | 47 | 0.95 | 1.33 | 1.80 | 2.01 | 2.31 | 3.13 | 3.69 | 4.57 |
| Acetone | 59 | 1.10 | 1.52 | 2.24 | 2.48 | 4.05 | 5.04 | 5.76 | 6.74 |
| Acetic acid | 61 | 0.95 | 1.38 | 2.11 | 2.42 | 2.65 | 4.02 | 5.26 | 6.22 |
| Isoprene | 69 | 0.41 | 0.60 | 1.05 | 1.07 | 1.92 | 2.56 | 3.25 | 3.91 |
| MVK+MACR | 71 | 0.13 | 0.21 | 0.41 | 0.43 | 0.79 | 1.05 | 1.35 | 1.58 |
| MEK | 73 | 0.28 | 0.41 | 0.59 | 0.67 | 1.00 | 1.29 | 1.52 | 1.82 |
| Benzene | 79 | 0.29 | 0.55 | 1.00 | 1.10 | 1.94 | 2.58 | 3.18 | 3.70 |
| Toluene | 93 | 0.94 | 1.34 | 2.34 | 2.72 | 4.23 | 5.49 | 6.25 | 7.06 |
| Xylenes | 107 | 0.43 | 0.83 | 1.69 | 1.94 | 3.32 | 4.28 | 5.09 | 5.68 |
| Monoterpenes | 137 | 0.15 | 0.23 | 0.43 | 0.46 | 0.87 | 1.22 | 1.37 | 1.71 |
| **Other species** | **Units** | | | | | | | | |
| Black carbon (BC) | $\mu g\ m^{-3}$ | 0.46 | 1.00 | 2.02 | 2.35 | 5.78 | 7.63 | 7.98 | 9.54 |
| Sulfur dioxide ($SO_2$) | ppb | 0.7 | 1.1 | 1.6 | 1.9 | 1.1 | 1.8 | 2.3 | 3.0 |
| Nitrogen oxides ($NO_x$) | ppb | 5.9 | 1.3 | 24.6 | 26.5 | 51.8 | 70.3 | 79.7 | 93.1 |
| Ozone ($O_3$) | ppb | 8.7 | 25.9 | 21.6 | 32.6 | 1.8 | 2.7 | 3.5 | 4.3 |
| Carbon monoxide (CO) | ppm | 0 | 0.1 | 0.3 | 0.4 | 1.0 | 1.2 | 1.4 | 1.7 |
| Carbon dioxide ($CO_2$) | ppm | 380 | 388 | 390 | 402 | 420 | 429 | 432 | 441 |





**Table 4:** Emission ratios and emission factors for biomass burning due to residential heating. Comparison with reported emission ratios from laboratory measurements from various biomass fuels types.

| Species | Emission Ratios ($\Delta$species/$\Delta$CO, ppb/ppm) | | | | Emission factors (g Kg$^{-1}$ of fuel) |
|---|---|---|---|---|---|
| | Residential heating (This work) | SW fuels (Warneke et al., 2011) | SE fuels (Warneke et al., 2011) | Pine spruce (Warneke et al., 2011) | Residential heating (This work) |
| Methanol | 14.1 ±2 | 3.75 | 9.47 | 8.49 | 0.38 ± 0.1 |
| Acetonitrile | 0.32 ± 0.05 | 0.56 | 1.03 | 1.05 | 0.01 ± 0.002 |
| Formic acid, Ethanol | 1.3 ± 0.3 | 0.77 | 1.08 | 1.0 | 0.045 ± 0.01 |
| Acetone | 2.9 ± 0.5 | 0.84 | 1.93 | 1.94 | 0.14 ± 0.02 |
| Acetic acid | 2.0 ± 1.0 | 4.84 | 13.61 | 8.19 | 0.11 ± 0.04 |
| Isoprene | 1.6 ± 0.22 | 0.53 | 1.38 | 1.57 | 0.09 ± 0.02 |
| MVK+MACR | 0.7 ± 0.1 | 0.43 | 1.08 | 1.32 | 0.04 ± 0.006 |
| MEK | 0.7 ± 0.2 | 0.41 | 1.28 | 1.17 | 0.04 ± 0.006 |
| Benzene | 1.8 ± 0.2 | 0.86 | 0.83 | 2.29 | 0.12 ± 0.01 |
| Toluene | 3.3 ± 0.8 | 0.30 | 0.48 | 0.81 | 0.26 ± 0.03 |
| Xylenes | 2.8 ± 0.6 | 0.19 | 0.35 | 0.60 | 0.26 ± 0.03 |
| C 9 Aromatics | 1.7 ± 0.4 | 0.11 | 0.32 | 0.48 | 0.18 ± 0.03 |
| C 10 Aromatics | 0.5 ± 0.1 | 0.10 | 0.22 | 0.33 | 0.06 ± 0.01 |
| Monoterpenes | 0.7 ± 0.2 | 0.16 | 0.55 | 1.25 | 0.08 ± 0.02 |
| $SO_2$ | 0.4 ± 0.5 | - | - | - | 0.02 ± 0.02 |
| $NO_x$ (as NO) | 52.7± 9 | - | - | - | 1.6 ± 0.2 |
| CO | - | - | - | - | 22.2 ± 2.8 |





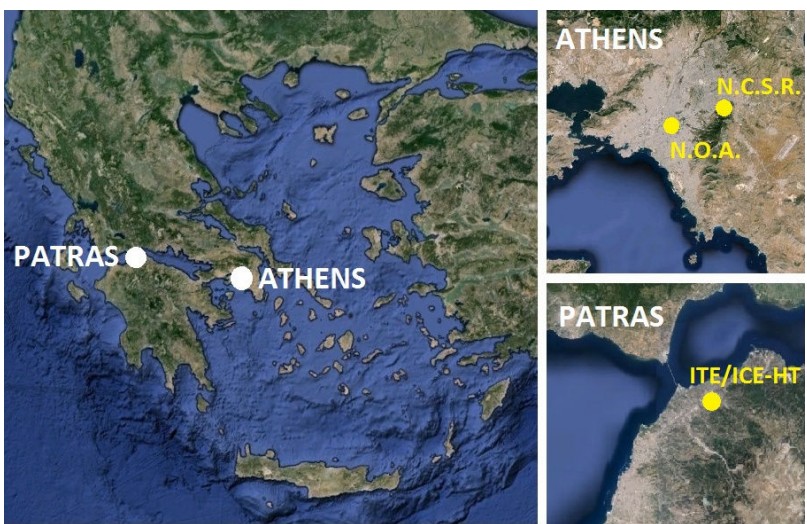

**Figure 1:** Maps of Greece, Athens and Patras together with the locations of the measurement sites used during the three campaigns.





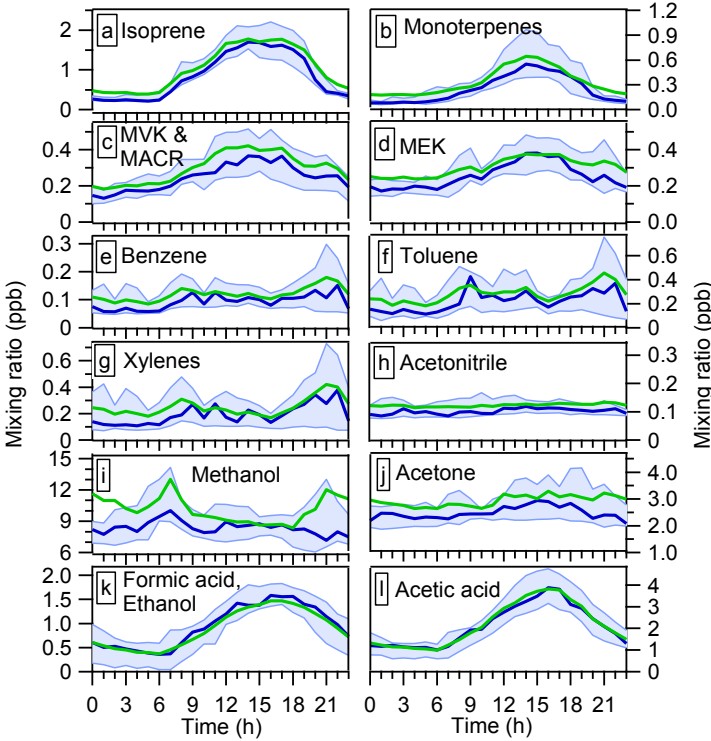

**Figure 2:** Average VOC diurnal profiles during the Patras summer campaign. Green lines present the average values. The median value is shown with blue. The blue area corresponds to the interquartile range.





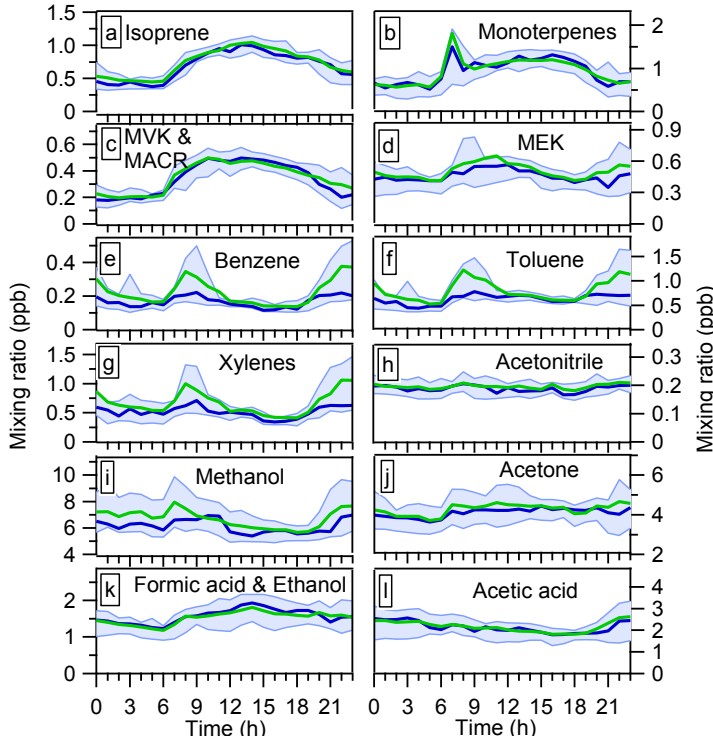

**Figure 3:** Average VOC diurnal profiles during the Athens summer campaign. Green lines present the average values. The median value is shown with blue. The blue area corresponds to the interquartile range.



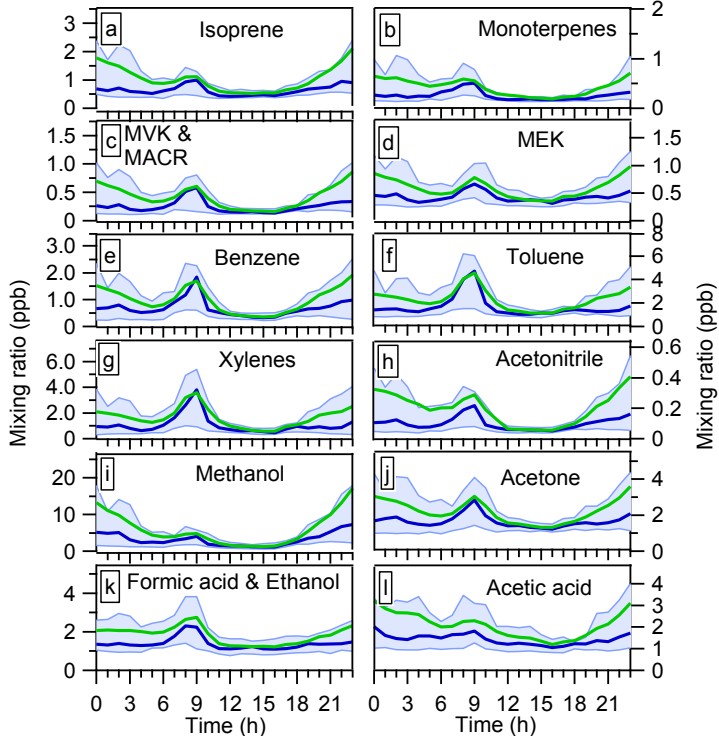

**Figure 4:** Average VOC diurnal profiles during the Athens winter campaign. Green lines present the average values. The median value is shown with blue. The blue area corresponds to the interquartile range.



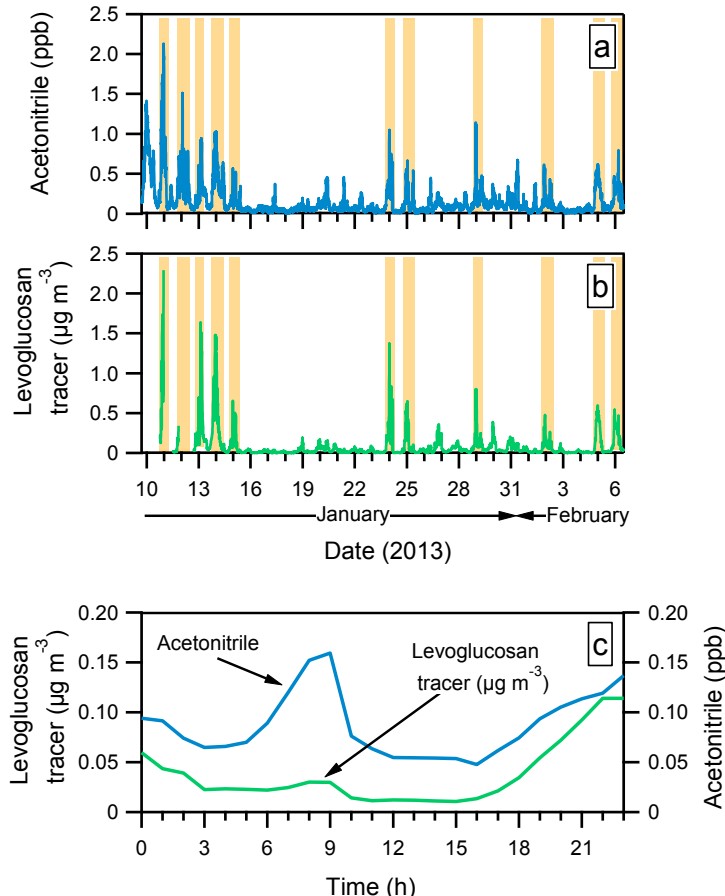

**Figure 5:** Time series of (a) AMS levoglucosan tracer (*m/z* 60 of the AMS) and (b) acetonitrile for the Athens winter campaign. Light brown indicates periods where wood burning contributed substantially to the total VOC and particulate levels. Also shown (c) their average diurnal profiles.




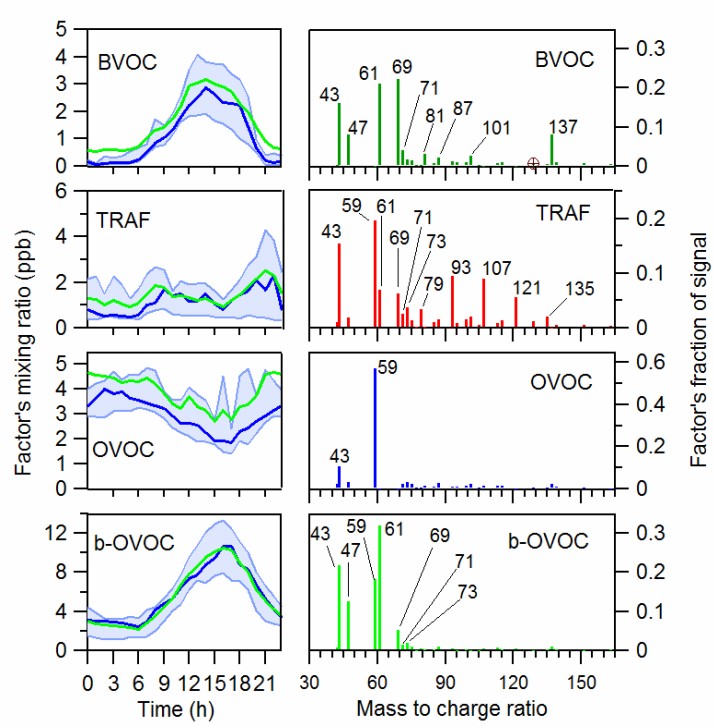

**Figure 6:** Results of Patras summer campaign PMF analysis. The left side shows the diurnal profiles of the factors. Green lines present the average values. The median value is shown with blue. The blue area is the interquartile range. The right side presents the *m/z* composition of each factor as a fraction of signal for each *m/z*.





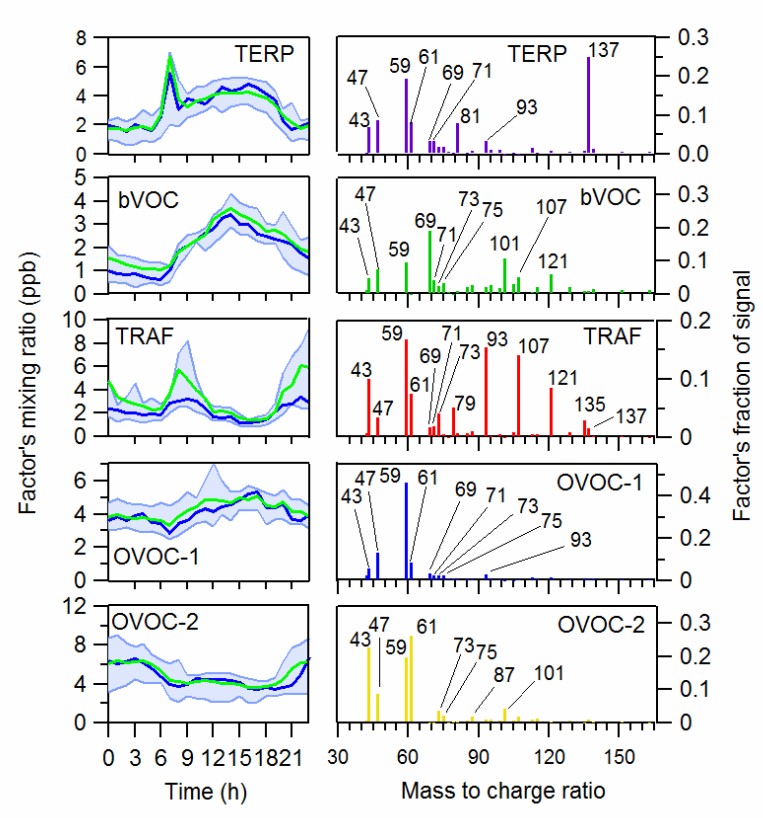

**Figure 7:** Results of Athens summer campaign PMF analysis. The left side shows the diurnal profiles of the factors. Green lines present the average values. The median value is shown with blue. The blue area is the interquartile range. The right side presents the *m/z* composition of each factor as a fraction of signal for each *m/z*.





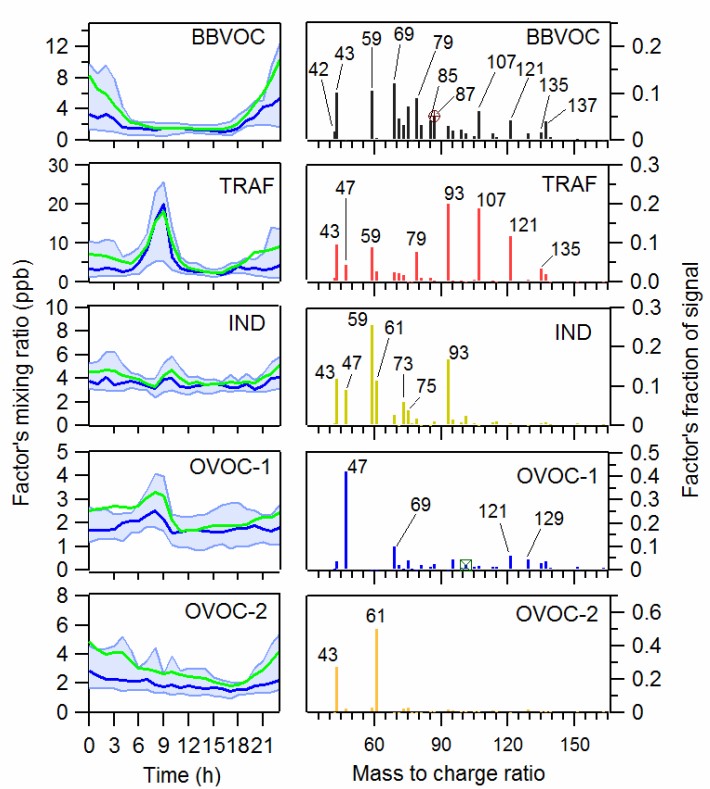

**Figure 8:** Results of Athens winter campaign PMF analysis. The left side shows the diurnal profiles of the factors. Green lines present the average values. The median value is shown with blue. The blue area is the interquartile range. The right side presents the *m/z* composition of each factor as a fraction of
5  signal for each *m/z*.