# Peer review of "Temporal variability and sources of VOCs in urban areas of Eastern Mediterranean"

_Atmospheric Chemistry and Physics, 2016_

## Referee Comment (RC1) · Anonymous Referee #1 · 13 Jun 2016

**General Comments:**

The paper by Kaltsonoudis et al. focuses on VOC composition measurements (performed using proton transfer reaction mass spectrometry) in Greece during the summer of 2012 and winter of 2013 and PMF source apportionment analyses using the measured datasets . The measurements in summer were conducted at two "background" sites in Patras (in June) and Athens (in July) during 2012, while the measurements in winter were conducted near the city centre of Athens in 2013.

The paper is written well and nicely structured. The reported VOC data and analyses will be useful for the atmospheric chemistry community. I recommend publishing the paper in ACP after the following major concerns have been addressed by the authors:

**Major Concerns:**

1) Experimental details: Section 2.0

While the authors have done a good job in providing an overview of the measurements, I find the current version "sketchy" when it comes to critical details pertaining to the PTR-MS measurements. The authors should provide additional information and clarify certain issues related to the measured datasets.

i) What was the residence time of air in the Teflon inlets during each of the PTR-MS deployments? What was the dwell time per m/z when the PTR-MS was used in the mass scan mode? The authors should specify the mass range intervals that were monitored in the 10 s measurement cycle.

ii) How often were the calibrations performed and did the sensitivities change over the course of the different deployments? Were humidity effects on the sensitivities of individual compounds considered during the calibration experiments? If not, how have these been addressed? The calculation of the concentrations for the calibrated compounds does not appear to have been normalized to the primary hydronium ion signal (that is cps/ppb appears to have been used instead of ncps /ppb). This could have a significant effect on the measurements depending on the difference in intensity of the primary ion signal between the calibration experiments and the ambient measurements. What are the detection limits for the compounds? The authors mention that for compounds where calibration was not possible, they used a value of $k = 2 \times 10^9 \, cm^3 \, s^{-1}$. They should also provide the Equation used by them in this case or cite the work listing the Equation at this place.

iii) It is mentioned that similar instrumentation was used for the winter deployment. Please clarify whether it was the same PTR-MS instrument?

iv) PTR-MS instrumental background measurements: It is mentioned that "blanks" were measured by passing the air through a Supelco activated charcoal filter. The more widely used

method in the measurement community is to use a heated Pt/Pd catalyst (or catalytic convertor) for generating the background air free of the relevant VOCs , as the activated charcoal filters are not good at removing oxygenated VOCs such as methanol. How did the authors ensure that they obtained good background measurements for such compounds (e.g. methanol). Note that humidity effects also affect the background of methanol strongly (see de Gouw and Warneke , 2007).

v) The authors acknowledge that the m/z =69 signal is a mix of furan and isoprene. The diurnal variability of the signal does indicate most of it is likely isoprene for the summertime deployments. Recent work using a high mass resolution PTR-TOF-MS instrument in the Kathmandu Valley (Sarkar et al., 2016) and previous laboratory measurements of biomass fuel emissions (Akagi et al. 2011) have shown that in air influenced by biomass burning, furan is still the minor contributor (25-30%) only. The authors may like to include and discuss the above works which support their contention.

2) PMF analyses: The authors should mention whether mass concentrations of VOCs were used as input for the PMF model. If so, were temperature and pressure corrections taken into account? A sentence providing the correlation between the estimated total VOC concentrations (PMF) Vs the observed total VOC concentrations (on x axis) should be added.

The PMF analyses should be improved by addition of conditional probability function analyses (see Fleming et al. 2012). This will yield potential wind direction dependence of sources and could be particularly useful for identifying strong point or area sources influencing the measurement site. During their summer deployments, the authors observed significantly higher methanol at one of the sites relative to the other. Some explanation regarding the possible cause/causes is required and this may help.

The inclusion of isoprene in the o-BVOC factor does not seem reasonable as being a hydrocarbon, isoprene cannot possibly be produced by atmospheric oxidation. The authors should clarify the same in the discussion.

I am somewhat surprised by the absence of fuel evaporation as a source. Considering previous studies in Paris and several other urban sites, it is important for the authors to discuss the absence of such a source.

Table 4: Authors may want to compare their ERs and EFs with values reported for other types of fuels (see Yokelson et al. 2013 and Stockwell et al. 2015) in addition to the Warneke et al. study already considered. This would enable them to include more fuel types for comparison with their ERs and EFs.

**Suggestions:**

While the authors have done a really good job in comparing their measurements to other studies in the Mediterranean region, a global perspective in terms of a comparison of the VOC concentration rankings with other sites in the world in summer (see for e.g. Table 3 of Sinha et al., 2014) and winter (see for e.g Figure 5 of Sarkar et al., 2016) would add a global flavor. A new Figure or Table is not necessary but a few sentences comparing the concentration ranking of VOCs observed during their deployments in Greece with similar recent measurements reported from some of the other urban /suburban sites elsewhere in the world would certainly help provide a more general perspective on the relative source strengths and ambient VOC composition in summer and winter, which is one of the primary objectives of their work. For example, it is interesting to note that the relative concentration ranking of VOCs in the Kathmandu valley in winter is quite different from that in Athens.

Discussion related to the Traffic source: A recent work published in GRL (see Link et al. 2016) reports that precursor compounds emitted from diesel vehicles can photo-chemically produce isocyanic acid (a compound recently reported by Chandra and Sinha, 2016 and Sarkar et al. 2016 using the PTR-MS technique in ambient field studies after the first field observations reported by Roberts et al., 2011 using negative ion CIMS). As the authors operated their PTR-MS in mass scan mode, it would be interesting to know if they observed any detectable signal at m/z 44 and whether the signal showed any diurnal variability. As an extremely toxic compound that can be measured using the PTR-MS, this information would certainly be useful to the readers.

**MINOR comments:**

Line 1; Page 5: Rate constants are normally reported in italics and small letters.. ..so it should be " $k$ " instead of "K".

Line 20, Page 6 : Why is the second traffic peak at 22:00 hr? This does not seem like the normal evening traffic rush hour period when office goers return home. Please clarify.

Line 30 Page 6: Benzene/Toluene ratio as a clock: The authors need to be careful here as some biomass sources can emit more benzene relative to toluene rendering the use of the T/B ratios as the sole criteria for aged air masses invalid, in particular when transported air masses across large distances are a strong possibility.

References:

Akagi,S.K.,Yokelson,R.J.,Wiedinmyer,C.,Alvarado,M.J.,Reid,J.S.,Karl,T.,Crounse,J.D.,andWennberg,P.O.:Emisionfactorsforopenanddomesticbiomassburningforuseinatmosphericmodels,Atmos.Chem.Phys.,11,4039–4072,doi:10.5194/acp-11-4039-2011,2011.

Chandra, B. P. and Sinha, V., Contribution of post-harvest agricultural paddy residue fires in the N.W. Indo-Gangetic Plain to ambient carcinogenic benzenoids, toxic isocyanic acid and carbon monoxide, Environment International, 88, 187-197, 2016.

de Gouw, J., andWarneke, C.: Measurements of volatile organic compounds in the Earth's atmosphere using proton-transfer-reaction mass spectrometry, Mass Spectrometry Reviews, 26, 223-257, doi:10.1002/mas.20119, 2007.

Fleming, Z. L., Monks, P. S., and Manning, A. J.: Review: Untangling the in
uence of airmass history in interpreting observed atmospheric composition, Atmospheric Research, 104105, 1-39, doi:10.1016/j.atmosres.2011.09.009, 2012.

Link, M. F., B. Friedman, R. Fulgham, P. Brophy, A. Galang, S. H. Jathar, P. Veres, J. M. Roberts, and D. K. Farmer (2016), Photochemical processing of diesel fuel emissions as a large secondary source of isocyanic acid (HNCO), Geophys. Res. Lett., 43, 4033–4041, doi:10.1002/2016GL068207

Roberts, J. M., Veres, P. R., Cochran, A. K., Warneke, C., Burling, I. R., Yokelson, R. J., Lerner, B., Gilman, J. B., Kuster, W. C., Fall, R., and de Gouw, J.: Isocyanic acid in the atmosphere and its possible link to smoke-related health e_ects, Proceedings of the National Academy of Sciences, doi:10.1073/pnas.1103352108, 2011.

Sarkar, C., Sinha, V., Kumar, V., Rupakheti, M., Panday, A., Mahata, K. S., Rupakheti, D., Kathayat, B., and Lawrence, M. G., Overview of VOC emissions and chemistry from PTR-TOF-MS measurements during the SusKat-ABC campaign: high acetaldehyde, isoprene and isocyanic acid in wintertime air of the Kathmandu Valley, Atmos. Chem. Phys., 16, 3979-4003, 2016.

Sinha,V., Kumar,V., and Sarkar,C.: Chemical composition of pre-monsoon air in the Indo–Gangetic Plain measured using a new PTR-MS and air quality facility: high surface ozone and strong influence of biomass burning, Atmos. Chem. Phys., 14, 5921-5941, 2014.

Stockwell, C. E., Veres, P. R., Williams, J., and Yokelson, R. J.: Characterization of biomass burning emissions from cooking , peat, crop residue, and other fuels with high-resolution proton-transfer-reaction time-of-flight mass spectrometry, Atmos. Chem. Phys., 15, 845-865, doi:10.5194/acp-15-845-2015, 2015.

Yokelson, R. J., Burling, I. R., Gilman, J. B., Warneke, C., Stockwell, C. E., de Gouw, J., Akagi, S. K., Urbanski, S. P., Veres, P., Roberts, J. M., Kuster, W. C., Reardon, J., Gri_th, D. W. T., Johnson, T. J., Hosseini, S., Miller, J. W., Cocker Iii, D. R., Jung, H., and Weise, D. R.: Coupling _eld and laboratory measurements to estimate the emission factors of identifed and unidentified trace gases for prescribed fires, Atmos. Chem. Phys., 13, 89-116, doi:10.5194/acp-13-89-2013, 2013.

---

## Referee Comment (RC2) · Anonymous Referee #2 · 17 Jun 2016

Review of "Temporal Variability and Souces of VOCs in Urban Areas of Eastern Mediterranean" by C. Kaltsonoudis et al. ACP-2016-358 Anonymous Referee Comments

This papers presents the results of VOC measurements using a PTRMS in two of the largest cities in Greece (Athens and Patra) during the summer and winter seasons. The paper presents a good overview of the most dominant VOCs observed and applies a standard PMF analysis to discuss the sources of VOCs. This is a well-organized and nicely written manuscript. These types of PMF analysis on gas phase measurements can be a powerful tool for understanding sources and is a generally underutilized tool. This paper provides a nice frame work for future PTR studies which will utilize PMF in a similar fashion.

[Figure]

General Comments

In generally I find the manuscript to be quite complete considering the extent of the data set collected. I have a few technical comments that will follow. My biggest comment is that a summary figure similar in design to Figure S20 would be highly beneficial in the main text. This paper ultimately focuses on using PMF to determine the impact of the various source on ambient VOCs. In this sense a summary figure using pie charts to summarize for each measurement location and season showing the relative impact of each factor would be a great way to relay the study's results in a compact manner that is easy to digest. The figures currently in the main text show the diurnal trends of the figures nicely, but that does nothing to show the relative impacts of each factor. Addition of a figure like this and a more distilled discussion of the results from that figure would really benefit to overall clarity of the analysis.

In general, there is a lack of discussion as to the potential for misidentification of observed m/z. The case that is most obvious is the treatment of isoprene, where the authors appropriately initially identify the potential for furan detection but end the discussion at that point. I would imagine that after performing PMF analysis and retrieving a BBVOC factor containing isoprene that the authors should discuss this 'isoprene' as potentially signal due entirely to furan, or at least partially. Continuing to label this measurement as isoprene is somewhat misleading, considering the potential overlap.

Section 2.2 on the experimental details is lacking some details on the method that are useful to the reader to understand the robustness of the measurements. Things like how long were the inlets overall in the various studies? Were the entire inlets heated to 60C or just a subsection where the PTRMS sampled. What has been done to correct the data for humidity effects? How was the normalization of the data done? How often were calibration performed?

Specific Comments

In the first sentence of the abstract, the phrase "urban background sites" is used. This

is not a commonly used term that requires some explanation, it may be best to simply state and urban site. Also in that first paragraph a sentence should be added introducing the winter season measurements. The last paragraph of the abstract launches into discussion of the results of winter measurements without first indicating that they were made.

Page 2, line 12, you need a new paragraph indent

Page 3, line 18, 'emissions' should be singular

Page 4, line 30, It rather well know that activated charcoal filters induce changes in humidity. As the sensitivity of the PTRMS to various species, such as methanol, is dependent on RH, how well do the authors believe they have been able to capture the true instrument background.

Page 7, line 14, the comment about elevated ozone levels being a result of long range transport seemly has no support in this manuscript. How do the authors know that? Were back trajectory analyses performed? This kind of a statement needs data to support it.

Page 7, line 33, suggest editing the phrase "peaked during noon at concentrations" with 'peaked at noon with concentrations"

Page 8, line 2, suggest reversing the order of 'significantly' and 'influenced'

Page 8, line 6, what spikes are being referred to here?

Page 8, line 23, suggest reversing the order of 'traffic' and 'hour'

Page 10, line 16-17, suggest rewriting to read "These periods (Table S2) occurred during the nighttime (18:00-06:00 LT) and were associated with acetonitrile…"

Page 10, Section 4, I would suggest that you reverse the order of the discussion in this section to first introduce the concept and equation for calculating EF, then discuss the results. Basically swap what is on page 11 with the text on page 10.

Page 11, line 5, suggest editing to read "For the CO2 emission factor (EFco2) a value of 1600..."

Page 11 line 27, why was the time period of two hours chosen here? Is there a basis for this time duration, was the correlation optimized at two hours, or is there a time shift that possibly induces a better correlation?

Page 12, line 8, Edit beginning of sentence to read "The OVOC factor..." Page 13, line 31, edit to read "Mixing ratios of OVOC-2 species were elevated during the night."

Page 15, line 1-4, That do you suggest that the OVOC-1 factor is? In the next paragraph you state that the OVOC-2 factor is associated with the background of VOCs but no discussion here.

Page16, line 12, Again to reiterate an above comment calling this isoprene in the biomass burning factor is likely misleading. If the author make a change here, an update to the manuscript tables is likely necessary.

Figures 2-4, I am not sure that the main text is the appropriate place for these figures. I would either move them into the supplemental or edit the figures to include only a few key species for all three measurement sites to compare the diurnals observed in each site/season.

---

## Author Comment (AC1) · 25 Aug 2016

*General Comments:*

**(1)** *The paper by Kaltsonoudis et al. focuses on VOC composition measurements (performed using proton transfer reaction mass spectrometry) in Greece during the summer of 2012 and winter of 2013 and PMF source apportionment analyses using the measured datasets. The measurements in summer were conducted at two "background" sites in Patras (in June) and Athens (in July) during 2012, while the measurements in winter were conducted near the city centre of Athens in 2013. The paper is written well and nicely structured. The reported VOC data and analyses will be useful for the atmospheric chemistry community. I recommend publishing the paper in ACP after the following major concerns have been addressed by the authors.*

[Figure]

We appreciate the constructive comments and suggestions by the referee. We have done our best to address all of them. Details are provided below.

*Major Concerns:*

**(2)** *Experimental details: Section 2.0. While the authors have done a good job in providing an overview of the measurements, I find the current version "sketchy" when it comes to critical details pertaining to the PTR-MS measurements. The authors should provide additional information and clarify certain issues related to the measured datasets.*

We have added the requested experimental information related to the PTR-MS measurements. Specific details are provided below.

**(i)** *What was the residence time of air in the Teflon inlets during each of the PTR-MS deployments? What was the dwell time per m/z when the PTR-MS was used in the mass scan mode? The authors should specify the mass range intervals that were monitored in the 10 s measurement cycle.*

The residence times for air through the sampling lines prior to the PTR-MS were 10 s for the Patras summer campaign, 12 s for the Athens summer campaign, and 16 s for the Athens winter campaign. The reference to the mass scan mode is a typographic error. The measurements were conducted in the multiple ion detection (MID) mode. The dwell times for these MID mode measurements were in the range of 5-500 ms. A 200 ms dwell time was used for most of the reported m/z values. For some of the higher m/z values a dwell time of 500 ms was selected. The dwell time for the m/z values 21, 30, 32 and 37 were 200, 5, 5 and 10 ms respectively in all three campaigns. The rest time (up to the 10 s value) were made up by the pauses (pause cal. function of the PTR-MS) during the m/z selections of the quadrupole. We have corrected the typo and added the above information in the revised manuscript.

**(ii)** *How often were the calibrations performed and did the sensitivities change over*

*the course of the different deployments? Were humidity effects on the sensitivities of individual compounds considered during the calibration experiments? If not, how have these been addressed? The calculation of the concentrations for the calibrated compounds does not appear to have been normalized to the primary hydronium ion signal (that is cps/ppb appears to have been used instead of ncps /ppb). This could have a significant effect on the measurements depending on the difference in intensity of the primary ion signal between the calibration experiments and the ambient measurements. What are the detection limits for the compounds? The authors mention that for compounds where calibration was not possible, they used a value of $k = 2 \times 10^9$ cm³ s⁻¹. They should also provide the Equation used by them in this case or cite the work listing the Equation at this place.*

Calibrations were performed once per week. The sensitivities for the reported compounds for all campaigns were in the range of 4.7 to 24 ncps. For example the sensitivity for acetonitrile for the three campaigns was 15.8-19.8 ncps, for benzene 9.5-13.3 ncps and for a-pinene 4.7-6.8 ncps. These values did not change significantly (less than 40 percent) during the measurements. The detection limits for the calibrated compounds based on Karl et al. (2003) were in the range of 34 to 97 ppt with the exception of methanol which ranged from 188 to 307 ppt for the three campaigns. Humidity effects on the sensitivities of the individual compounds were not considered during the calibrations. A precision calibrator (Teledyne, model 702) was used for the dilution of the VOC standard with VOC free air (using a Supelco filter). This calibrator type does not make adjustments for the humidity. The ratio of m/z 37 to m/z 19 was $0.039 \pm 0.009$ during the Patras summer campaign, $0.044 \pm 0.012$ during the Athens summer campaign, and $0.038 \pm 0.008$ during the Athens winter campaign. In all campaigns the m/z 37 to m/z 19 ratio was always less than 0.06 (typically ranging from 0.025 to 0.05). This ratio is considered low and stable, thus no corrections were applied for the $H_3O^+(H_2O)$ ion. The concentrations of all compounds were normalized to the primary hydronium ion signal (please see Equations 1 and 2 of the revised manuscript). For compounds for which calibration was not possible, the corresponding mixing ratios were estimated

based on Equation 3 of Taipale et al. (2008). The above additional information and equations have been added to the revised manuscript.

**(iii)** *It is mentioned that similar instrumentation was used for the winter deployment. Please clarify whether it was the same PTR-MS instrument?*

It was the same PTR-MS instrument. This is now clarified in the revised paper.

**(iv)** *PTR-MS instrumental background measurements: It is mentioned that "blanks" were measured by passing the air through a Supelco activated charcoal filter. The more widely used method in the measurement community is to use a heated Pt/Pd catalyst (or catalytic convertor) for generating the background air free of the relevant VOCs, as the activated charcoal filters are not good at removing oxygenated VOCs such as methanol. How did the authors ensure that they obtained good background measurements for such compounds (e.g. methanol). Note that humidity effects also affect the background of methanol strongly (see de Gouw and Warneke, 2007).*

This is a valid concern. Unfortunately a catalytic convertor was not available for generating VOC-free air. Instead the activated charcoal filter (Supelco) was used as the final purification step of the "clean" air. This may not have been sufficient to generate methanol - free air and could have added additional uncertainty to our methanol measurements. Given that the methanol measurements are of secondary importance for the purposes of this manuscript we have not included them in the revised paper.

**(v)** *The authors acknowledge that the m/z =69 signal is a mix of furan and isoprene. The diurnal variability of the signal does indicate most of it is likely isoprene for the summertime deployments. Recent work using a high mass resolution PTR-TOF-MS instrument in the Kathmandu Valley (Sarkar et al., 2016) and previous laboratory measurements of biomass fuel emissions (Akagi et al. 2011) have shown that in air influenced by biomass burning, furan is still the minor contributor (25-30 percent) only. The*

*authors may like to include and discuss the above works which support their contention.*

This is a good suggestion. Since the quadrupole of the PTR-MS cannot distinguish between the different species in several m/z values (including m/z 69) we have chosen to report most of the common atmospheric compounds detected in the corresponding m/z values. For the summer campaigns the m/z 69 diurnal profile indicates that isoprene is the dominant molecule for this m/z. For the Athens winter campaign where biomass burning had a strong influence on the reported concentrations of m/z 69 we both report the corresponding atmospheric compounds and also comment that isoprene is the predominant compound for m/z 69. We have added discussion of this point in the revised paper together with the corresponding references suggested by the reviewer.

**(3)** *PMF analyses: The authors should mention whether mass concentrations of VOCs were used as input for the PMF model. If so, were temperature and pressure corrections taken into account? A sentence providing the correlation between the estimated total VOC concentrations (PMF) vs the observed total VOC concentrations (on x axis) should be added.*

Mixing ratios (ppb) were used as the input for the PMF model and therefore temperature and pressure corrections were not applied. We have added a figure (Figure S47) in the Supplementary Material showing the correlation between the total VOC concentrations estimated by the PMF model versus the measured ones for the three campaigns. The estimated concentrations of the PMF model correlated well ($R^2$ ranged from 0.994 to 0.999) with the measured values for the three campaigns. We have added a sentence with the correlation results and the corresponding figure to the revised manuscript.

**(4)** *The PMF analyses should be improved by addition of conditional probability function analyses (see Fleming et al. 2012). This will yield potential wind direction dependence of sources and could be particularly useful for identifying strong point or area sources*

*influencing the measurement site. During their summer deployments, the authors ob-
served significantly higher methanol at one of the sites relative to the other. Some
explanation regarding the possible cause/causes is required and this may help.*

The origin of the air masses during the three campaigns has been investigated using
FLEXPART and HYSPLIT. These data are available in the SI of Kostenidou et al. (2015)
for the summer measurements in Patras and Athens and in Florou et al. (2016) for the
winter measurements. The dependence of the PMF Factors on wind direction has been
also investigated with rose plots. The corresponding rose plots have been included in
the revised SI. We have added a paragraph discussing these results in the revised
paper.

**(5)** *The inclusion of isoprene in the o-BVOC factor does not seem reasonable as being
a hydrocarbon, isoprene cannot possibly be produced by atmospheric oxidation. The
authors should clarify the same in the discussion.*

We clarify in the revised paper that the b-OVOC factor for the Patras summer campaign
includes mainly products of the oxidation of biogenic VOCs, but also some primary
VOCs. Approximately 70 percent of the measured isoprene during the Patras summer
campaign was included by the PMF analysis in the BVOC factor (see also Figure S20
in the SI). The remaining 30 percent was included in other factors, with 15 percent
in the b-OVOC factor. This small amount of isoprene assigned to this factor can be
justified either as direct biogenic emissions that have originated far from the area close
to the measurement site and have not yet reacted or it can be due to the uncertainties
of the PMF analysis. The factor was named b-OVOC (biogenic oxygenated VOCs) due
to the strong influence by formic acid, acetone and acetic acid. We have added the
above clarifications to the paper.

**(6)** *I am somewhat surprised by the absence of fuel evaporation as a source. Con-
sidering previous studies in Paris and several other urban sites, it is important for the*

[Figure]

*authors to discuss the absence of such a source.*

Most of the urban source apportionment VOC studies have separated the anthropogenic VOCs into several categories including fuel evaporation, industrial solvents, diesel exhaust emissions, etc. (Brown et al., 2007; Tolga et al., 2007; Badol et al., 2008; Leuchner and Rappengluck, 2010; Yurdakul et al., 2013; Boynard et al., 2014; Wang et al., 2014). In the present study the PMF analysis did not produce such factors since light alkanes were not measured, in contrast to the previous studies based on gas chromatography. The vehicular exhaust and LPG sources widely reported in the literature include a large fraction of these species, thus such a separation was not favored by the PMF analysis. We have included this important point in a new paragraph synthesizing the results of the PMF analysis.

**(7)** *Table 4: Authors may want to compare their ERs and EFs with values reported for other types of fuels (see Yokelson et al. 2013 and Stockwell et al. 2015) in addition to the Warneke et al. study already considered. This would enable them to include more fuel types for comparison with their ERs and EFs.*

We have followed the reviewer's suggestion and added the comparisons with the results of Yogelson et al. (2013) and Stockwell et al. (2015). Table 4 has been replaced by two new tables (one for emission ratios and one for emission factors) in the revised paper.

*Suggestions:*

**(8)** *While the authors have done a really good job in comparing their measurements to other studies in the Mediterranean region, a global perspective in terms of a comparison of the VOC concentration rankings with other sites in the world in summer (see for e.g. Table 3 of Sinha et al., 2014) and winter (see for e.g Figure 5 of Sarkar et al., 2016) would add a global flavor. A new Figure or Table is not necessary but a few sentences comparing the concentration ranking of VOCs observed during their*

*deployments in Greece with similar recent measurements reported from some of the other urban /suburban sites elsewhere in the world would certainly help provide a more general perspective on the relative source strengths and ambient VOC composition in summer and winter, which is one of the primary objectives of their work. For example, it is interesting to note that the relative concentration ranking of VOCs in the Kathmandu valley in winter is quite different from that in Athens.*

We have extended our comparison to include other areas and comparisons with the results of Sinha et al. (2014) and Sarkar et al. (2016).

**(9)** *Discussion related to the Traffic source: A recent work published in GRL (see Link et al. 2016) reports that precursor compounds emitted from diesel vehicles can photo-chemically produce isocyanic acid (a compound recently reported by Chandra and Sinha, 2016 and Sarkar et al. 2016 using the PTR-MS technique in ambient field studies after the first field observations reported by Roberts et al., 2011 using negative ion CIMS). As the authors operated their PTR-MS in mass scan mode, it would be interesting to know if they observed any detectable signal at m/z 44 and whether the signal showed any diurnal variability. As an extremely toxic compound that can be measured using the PTR-MS, this information would certainly be useful to the readers.*

Unfortunately we only have limited mass scan mode measurements (see also our response to Comment 2i). During each campaign we used the mass scan mode for 10-15 min every 2-3 days in order to take a look beyond the normally measured m/z values. These data were not sufficient to reach any robust conclusion regarding isocyanic acid. For example, a mass scan measurement during the morning rush hour (10:00) in Athens during the winter gave concentrations for m/z 44 lower than 60-80 ppt, while there was 1.7 ppb of benzene and 3.9 ppb of toluene during the same period.

*Minor comments:*

**(10)** *Line 1; Page 5: Rate constants are normally reported in italics and small letters..*

*..so it should be " k" instead of "K".*

We have replaced "K" with "k".

**(11)** *Line 20, Page 6: Why is the second traffic peak at 22:00 hr? This does not seem like the normal evening traffic rush hour period when office goers return home. Please clarify.*

This evening traffic peak is a little after the closing of the various merchant shops in the summertime and is consistent with the increased traffic observed during these periods as both the customers and owners return home. Kostenidou et al. (2015) observed HOA and BC peaks during the same period in both Athens and Patras. We have added an explanation of this peak to the revised paper.

**(12)** *Line 30 Page 6: Benzene/Toluene ratio as a clock: The authors need to be careful here as some biomass sources can emit more benzene relative to toluene rendering the use of the T/B ratios as the sole criteria for aged air masses invalid, in particular when transported air masses across large distances are a strong possibility.*

During the Patras summer campaign we could not identify any significant biomass burning sources either close to the measurement site or transported from larger distances. The acetonitrile levels were low and there were no significant peaks indicating biomass buring sources. Additionally the AMS measurements (Kostenidou et al., 2015) did not show any biomass burning related factor in the PMF analysis. We have added however a qualifier that this analysis assumes no influence from biomass burning sources.

**References**

Akagi, S. K., Yokelson, R. J., Wiedinmyer, C., Alvarado, M. J., Reid, J. S., Karl, T., Crounse, J. D., and Wennberg, P. O.: Emision factors for open and domestic

biomass burning for use in atmospheric models, Atmos. Chem. Phys., 11, 4039–4072, doi:10.5194/acp-11-4039-2011, 2011.

Badol, C., Locoge, N., and Galloo, J.: Using a source-receptor approach to characterize VOC behavior in a French urban area influenced by industrial emissions Part II: Source contribution assessment using the Chemical Mass Balance (CMB) model. Science of the Total Environment. 389, 429-440, 2008.

Boynard, A., Borbon, A., Leonardis, T., Barletta, B., Meinardi, S., Blake, D., and Locoge, N.: Spatial and seasonal variability of measured anthropogenic nonmethane hydrocarbons in urban atmospheres: Implication on emission ratios. Atmospheric Environment, 82, 258-267, 2014.

Brown, S., Frankel, A. And Hafner, H.: Source apportionment of VOCs in the Los Angeles area using positive matrix factorization. Atmospheric Environment, 41, 227–237, 2007.

Chandra, B. P. and Sinha, V., Contribution of post-harvest agricultural paddy residue fires in the N.W. Indo-Gangetic Plain to ambient carcinogenic benzenoids, toxic isocyanic acid and carbon monoxide, Environment International, 88, 187-197, 2016.

DeGouw, J., and Warneke, C.: Measurements of volatile organic compounds in the Earth's atmosphere using proton-transfer-reaction mass spectrometry, Mass Spectrometry Reviews, 26, 223-257, 2007.

Fleming, Z. L., Monks, P. S., and Manning, A. J.: Review: Untangling the influence of airmass history in interpreting observed atmospheric composition, Atmospheric Research, 104105, 1-39, doi:10.1016/j.atmosres.2011.09.009, 2012.

Karl, T., Jobson, T., Kuster, W.C.,Williams, E., Stutz, J., Shetter, R., Hall, S.R., Goldan, P., Fehsenfeld, F. and Lindinger, W.: Use of proton-transfer-reaction mass spectrometry to characterize volatile organic compound sources at the La Porte super site during the Texas Air Quality Study 2000. J. Geophysical Res. 108, 4508,

doi:10.1029/2002JD003333, 2003.

Karl, T., Christian, T., Yokelson, R., Artaxo, P., Hao W. and Guenther, A.: The Tropical Forest and Fire Emissions Experiment: method evaluation of volatile organic compound emissions measured by PTR-MS, FTIR, and GC from tropical biomass burning. Atmos. Chem. Phys., 7, 5883–5897, 2007.

Kostenidou, E., Florou, K., Kaltsonoudis, C., Tsiflikiotou, M., Eleftheriadis, K. and Pandis, S.N.: Sources and chemical characterization of organic aerosol during the summer in the eastern Mediterranean. Atmos. Chem. Phys., 15, 11355-11371, 2015.

Kouvarakis, G., Vrekoussis, M., Michalopoulos, N., Kourtidis, K. and Rappenglueck B.: Spatial and temporal variability of tropospheric ozone (O3) in the boundary layer above the Aegean Sea (eastern Mediterranean). J. of Geophysical Res., 107, 8137, doi:10.1029/2000JD000081, 2002.

Leuchner, M., and Rappengluck, B.: VOC source–receptor relationships in Houston during TexAQS-II. Atmospheric Environment, 44, 4056–4067, 2010.

Lindinger, W., Hansel, A. and Jordan, A.: On-line monitoring of volatile organic compounds at pptv levels by means of Proton-Transfer-Reaction Mass Spectrometry (PTR-MS) Medical applications, food control and environmental research. International Journal of Mass Spectrometry and Ion Processes. 173, 191-241, 1998.

Link, M. F., B. Friedman, R. Fulgham, P. Brophy, A. Galang, S. H. Jathar, P. Veres, J. M. Roberts, and D. K. Farmer (2016), Photochemical processing of diesel fuel emissions as a large secondary source of isocyanic acid (HNCO), Geophys. Res. Lett., 43, 4033–4041, doi:10.1002/2016GL068207.

Roberts, J. M., Veres, P. R., Cochran, A. K., Warneke, C., Burling, I. R., Yokelson, R. J., Lerner, B., Gilman, J. B., Kuster, W. C., Fall, R., and de Gouw, J.: Isocyanic acid in the atmosphere and its possible link to smoke-related health effects, Proceedings of the National Academy of Sciences, doi:10.1073/pnas.1103352108, 2011.

Sarkar, C., Sinha, V., Kumar, V., Rupakheti, M., Panday, A., Mahata, K. S., Rupakheti, D., Kathayat, B., and Lawrence, M. G., Overview of VOC emissions and chemistry from PTR-TOF-MS measurements during the SusKat-ABC campaign: high acetaldehyde, isoprene and isocyanic acid in wintertime air of the Kathmandu Valley, Atmos. Chem. Phys., 16, 3979-4003, 2016.

Sinha,V., Kumar,V., and Sarkar,C.: Chemical composition of pre-monsoon air in the Indo–Gangetic Plain measured using a new PTR-MS and air quality facility: high surface ozone and strong influence of biomass burning, Atmos. Chem. Phys., 14, 5921-5941, 2014.

Stockwell, C. E., Veres, P. R., Williams, J., and Yokelson, R. J.: Characterization of biomass burning emissions from cooking , peat, crop residue, and other fuels with high-resolution proton-transfer-reaction time-of-flight mass spectrometry, Atmos. Chem. Phys., 15, 845-865, doi:10.5194/acp-15-845-2015, 2015.

Taipale, R., Ruuskanen, T.M., Rinne, J., Kajos, M.K., Hakola, H., Pohja, T., and Kulmala, M.: Technical Note: Quantitative long-term measurements of VOC concentrations by PTR-MS – measurement, calibration, and volume mixing ratio calculation methods. Atmos. Chem. Phys., 8, 6681–6698, 2008.

Tolga, E., Cetin, B., Cetin, E., Bayram, A., and Odabasi, M.: Characterization of volatile organic compounds (VOCs) and their sources in the air of Izmir, Turkey. Environ. Monit. Assess. 133, 149–160, 2007.

Yokelson, R. J., Burling, I. R., Gilman, J. B., Warneke, C., Stockwell, C. E., de Gouw, J., Akagi, S. K., Urbanski, S. P., Veres, P., Roberts, J. M., Kuster, W. C., Reardon, J., Griffith, D. W. T., Johnson, T. J., Hosseini, S., Miller, J. W., Cocker Iii, D. R., Jung, H., and Weise, D. R.: Coupling field and laboratory measurements to estimate the emission factors of identified and unidentified trace gases for prescribed fires, Atmos. Chem. Phys., 13, 89-116, doi:10.5194/acp-13-89-2013, 2013.

**[ACPD](javascript:void(0))**

Interactive
comment
Yurdakul, S., Civan, M., and Tuncel, G.: Volatile organic compounds in suburban Ankara atmosphere, Turkey: Sources and variability. Atmospheric Research, 120, 298–311, 2013.

Wang, M., Shao, M., Chen, W., Yuan, B., Lu, S., Zhang, Q., Zeng, L., and Wang, Q.: A temporally and spatially resolved validation of emission inventories by measurements of ambient volatile organic compounds in Beijing, China. Atmos. Chem. Phys., 14, 5871–5891, 2014.

---

## Author Comment (AC2) · 25 Aug 2016

*This papers presents the results of VOC measurements using a PTRMS in two of the largest cities in Greece (Athens and Patra) during the summer and winter seasons. The paper presents a good overview of the most dominant VOCs observed and applies a standard PMF analysis to discuss the sources of VOCs. This is a well-organized and nicely written manuscript. These types of PMF analysis on gas phase measurements can be a powerful tool for understanding sources and is a generally underutilized tool. This paper provides a nice frame work for future PTR studies which will utilize PMF in a similar fashion.*

*General Comments*

**(1)** *In generally I find the manuscript to be quite complete considering the extent of*

[Figure]

*the data set collected. I have a few technical comments that will follow. My biggest comment is that a summary figure similar in design to Figure S20 would be highly beneficial in the main text. This paper ultimately focuses on using PMF to determine the impact of the various sources on ambient VOCs. In this sense a summary figure using pie charts to summarize for each measurement location and season showing the relative impact of each factor would be a great way to relay the study's results in a compact manner that is easy to digest. The figures currently in the main text show the diurnal trends of the figures nicely, but that does nothing to show the relative impacts of each factor. Addition of a figure like this and a more distilled discussion of the results from that figure would really benefit to overall clarity of the analysis.*

We have followed the reviewer's suggestion and transferred this figure summarizing the PMF results from the SI to the revised paper.

**(2)** *In general, there is a lack of discussion as to the potential for misidentification of observed m/z. The case that is most obvious is the treatment of isoprene, where the authors appropriately initially identify the potential for furan detection but end the discussion at that point. I would imagine that after performing PMF analysis and retrieving a BBVOC factor containing isoprene that the authors should discuss this "isoprene" as potentially signal due entirely to furan, or at least partially. Continuing to label this measurement as isoprene is somewhat misleading, considering the potential overlap.*

A similar point was made by Reviewer 1 (see Comment 2v). We have added discussion of the potential contributions of isoprene and furan to the signal at m/z 69 arguing that it is mostly isoprene in all of our data sets based on its diurnal profile during the summer and the biomass burning VOC literature (citations have been added) during the winter. After this discussion, our use of the term isoprene is justified but we still point out the potential small interference by furan.

**(3)** *Section 2.2 on the experimental details is lacking some details on the method that*

*are useful to the reader to understand the robustness of the measurements. Things like how long were the inlets overall in the various studies? Were the entire inlets heated to 60 C or just a subsection where the PTRMS sampled. What has been done to correct the data for humidity effects? How was the normalization of the data done? How often were calibration performed?*

A subsection (approximately 1 m) of the inlet live was heated to 60 C. Sampling lines were 6-10 m long. The residence times for air through the sampling lines prior to the PTR-MS were 10 s for the Patras summer campaign, 12 s for the Athens summer campaign, and 16 s for the Athens winter campaign. Calibrations were performed once per week. The sensitivities for the reported compounds for all campaigns were in the range of 4.7 to 24 ncps. For example the sensitivity for acetonitrile for the three campaigns was 15.8-19.8 ncps, for benzene 9.5-13.3 ncps and for a-pinene 4.7-6.8 ncps. These values did not change significantly (less than 40 percent) during the measurements. The detection limits for the calibrated compounds based on Karl et al. (2003) were in the range of 34 to 97 ppt with the exception of methanol which ranged from 188 to 307 ppt for the three campaigns. Humidity effects on the sensitivities of the individual compounds were not considered during the calibrations. A precision calibrator (Teledyne, model 702) was used for the dilution of the VOC standard with VOC free air (using a Supelco filter). This calibrator type does not make adjustments for the humidity. The ratio of m/z 37 to m/z 19 was $0.039 \pm 0.009$ during the Patras summer campaign, $0.044 \pm 0.012$ during the Athens summer campaign, and $0.038 \pm 0.008$ during the Athens winter campaign. In all campaigns the m/z 37 to m/z 19 ratio was always less than 0.06 (typically ranging from 0.025 to 0.05). This ratio is considered low and stable, thus no corrections were applied for the $H_3O^+(H_2O)$ ion. The concentrations of all compounds were normalized to the primary hydronium ion signal (please see Equations 1 and 2 of the revised manuscript). For compounds for which calibration was not possible, the corresponding mixing ratios were estimated based on Equation 3 of Taipale et al. (2008). The above additional information and equations have been added to the revised manuscript.

[Figure]

*Specific Comments*

**(4)** *In the first sentence of the abstract, the phrase "urban background sites" is used. This is not a commonly used term that requires some explanation. It may be best to simply state and urban site. Also in that first paragraph a sentence should be added introducing the winter season measurements. The last paragraph of the abstract launches into discussion of the results of winter measurements without first indicating that they were made.*

We have deleted the term "background" given that the lack of major nearby sources (e.g., roads) is also mentioned in the site description. The last paragraph of the abstract has been rewritten explaining first when and where the measurements took place.

**(5)** *Page 2, line 12, you need a new paragraph indent.*

A new paragraph indent has been added at this point.

**(6)** *Page 3, line 18, 'emissions' should be singular.*

We have corrected the typo.

**(7)** *Page 4, line 30. It is rather well known that activated charcoal filters induce changes in humidity. As the sensitivity of the PTRMS to various species, such as methanol, is dependent on RH, how well do the authors believe they have been able to capture the true instrument background?*

This point was also raised by Referee 1 (see Comment 2iv). Unfortunately a catalytic convertor was not available for generating VOC-free air. Instead the activated charcoal filter (Supelco) was used as the final purification step of the "clean" air. This may not have been sufficient to generate methanol - free air and could have added additional uncertainty to our methanol measurements. Given that the methanol measurements

are of secondary importance for the purposes of this manuscript we have not included them in the revised paper.

**(8)** *Page 7, line 14, the comment about elevated ozone levels being a result of long range transport seemly has no support in this manuscript. How do the authors know that? Were back trajectory analyses performed? This kind of a statement needs data to support it.*

The importance of the long range transport can be seen in Figure S3 of the revised paper showing the ozone concentrations during this windy period. The maximum ozone concentration was actually observed a little before the midnight of June 18. This was actually the highest ozone level during the whole measurement period. This was in contrast to the low levels of $NO_x$ and BC observed during the same period. These results shown in Figure S3 strongly support the conclusion that the high levels of ozone were due to long range transport. Similar conclusions have been reached by Kouvarakis et al. (2002) reporting measurements performed onboard a cruise ship travelling on a regular basis in the area. They concluded that long range transport is the main factor contribution to high ozone levels in eastern Greece. We have added the above arguments and the reference to the work Kouvarakis et al. (2002) in the revised paper.

**(9)** *Page 7, line 33, suggest editing the phrase "peaked during noon at concentrations" with 'peaked at noon with concentrations"*

We have made the suggested correction.

**(10)** *Page 8, line 2, suggest reversing the order of 'significantly' and 'influenced'.*

We have made the suggested correction.

**(11)** *Page 8, line 6, what spikes are being referred to here?*

This refers to the elevated monoterpene concentrations at around 7:00 seen in Figure 3b. We have replaced "spikes" with "elevated concentrations" to avoid confusion.

**(12)** *Page 8, line 23, suggest reversing the order of 'traffic' and 'hour'.*

We have made the change.

**(13)** *Page 10, line 16-17, suggest rewriting to read "These periods (Table S2) occurred during the nighttime (18:00-06:00 LT) and were associated with acetonitrile".*

We have rewritten this sentence.

**(14)** *Page 10, Section 4, I would suggest that you reverse the order of the discussion in this section to first introduce the concept and equation for calculating EF, then discuss the results. Basically swap what is on page 11 with the text on page 10.*

We have followed the suggestion of the reviewer and changed the order of these sections.

**(15)** *Page 11, line 5, suggest editing to read "For the CO2 emission factor (EFCO2) a value of 1600".*

We have made this suggested change.

**(16)** *Page 11 line 27, why was the time period of two hours chosen here? Is there a basis for this time duration, was the correlation optimized at two hours, or is there a time shift that possibly induces a better correlation?*

The correlation was optimized at two hours. We have added text here to clarify this point.

[Figure]
**Interactive comment**

**(17)** *Page 12, line 8, Edit beginning of sentence to read "The OVOC factor".*

Done.

**(18)** *Page 13, line 31, edit to read "Mixing ratios of OVOC-2 species were elevated during the night."*

We have rephrased this sentence.

**(19)** *Page 15, line 1-4, That do you suggest that the OVOC-1 factor is? In the next paragraph you state that the OVOC-2 factor is associated with the background of VOCs but no discussion here.*

Factor OVOC-1 in the Athens winter campaign could not be associated with any specific potential source. We have added this explanation in the paper.

**(20)** *Page16, line 12, Again to reiterate an above comment calling this isoprene in the biomass burning factor is likely misleading. If the authors make a change here, an update to the manuscript tables is likely necessary.*

We have replaced "isoprene" with "m/z 69" at this point. The same has been made in the section discussing the wintertime measurements in Athens.

**(21)** *Figures 2-4, I am not sure that the main text is the appropriate place for these figures. I would either move them into the supplemental or edit the figures to include only a few key species for all three measurement sites to compare the diurnals observed in each site/season.*

We believe that these figures are quite useful for the reader of the paper as they summarize the behavior of some of the major species. We have followed the reviewer's suggestion and removed the methanol and formic acid diurnal profiles from these three figures in the revised paper.

**References**

Akagi, S. K., Yokelson, R. J., Wiedinmyer, C., Alvarado, M. J., Reid, J. S., Karl, T., Crounse, J. D., and Wennberg, P. O.: Emision factors for open and domestic biomass burning for use in atmospheric models, Atmos. Chem. Phys., 11, 4039–4072, doi:10.5194/acp-11-4039-2011, 2011.

Badol, C., Locoge, N., and Galloo, J.: Using a source-receptor approach to characterize VOC behavior in a French urban area influenced by industrial emissions Part II: Source contribution assessment using the Chemical Mass Balance (CMB) model. Science of the Total Environment. 389, 429-440, 2008.

Boynard, A., Borbon, A., Leonardis, T., Barletta, B., Meinardi, S., Blake, D., and Locoge, N.: Spatial and seasonal variability of measured anthropogenic nonmethane hydrocarbons in urban atmospheres: Implication on emission ratios. Atmospheric Environment, 82, 258-267, 2014.

Brown, S., Frankel, A. And Hafner, H.: Source apportionment of VOCs in the Los Angeles area using positive matrix factorization. Atmospheric Environment, 41, 227–237, 2007.

Chandra, B. P. and Sinha, V., Contribution of post-harvest agricultural paddy residue fires in the N.W. Indo-Gangetic Plain to ambient carcinogenic benzenoids, toxic isocyanic acid and carbon monoxide, Environment International, 88, 187-197, 2016.

DeGouw, J., and Warneke, C.: Measurements of volatile organic compounds in the Earth's atmosphere using proton-transfer-reaction mass spectrometry, Mass Spectrometry Reviews, 26, 223-257, 2007.

Fleming, Z. L., Monks, P. S., and Manning, A. J.: Review: Untangling the influence of airmass history in interpreting observed atmospheric composition, Atmospheric Research, 104105, 1-39, doi:10.1016/j.atmosres.2011.09.009, 2012.

Karl, T., Jobson, T., Kuster, W.C.,Williams, E., Stutz, J., Shetter, R., Hall, S.R.,

Goldan, P., Fehsenfeld, F. and Lindinger, W.: Use of proton-transfer-reaction mass spectrometry to characterize volatile organic compound sources at the La Porte super site during the Texas Air Quality Study 2000. J. Geophysical Res. 108, 4508, doi:10.1029/2002JD003333, 2003.

Karl, T., Christian, T., Yokelson, R., Artaxo, P., Hao W. and Guenther, A.: The Tropical Forest and Fire Emissions Experiment: method evaluation of volatile organic compound emissions measured by PTR-MS, FTIR, and GC from tropical biomass burning. Atmos. Chem. Phys., 7, 5883–5897, 2007.

Kostenidou, E., Florou, K., Kaltsonoudis, C., Tsiflikiotou, M., Eleftheriadis, K. and Pandis, S.N.: Sources and chemical characterization of organic aerosol during the summer in the eastern Mediterranean. Atmos. Chem. Phys., 15, 11355-11371, 2015.

Kouvarakis, G., Vrekoussis, M., Michalopoulos, N., Kourtidis, K. and Rappenglueck B.: Spatial and temporal variability of tropospheric ozone (O3) in the boundary layer above the Aegean Sea (eastern Mediterranean). J. of Geophysical Res., 107, 8137, doi:10.1029/2000JD000081, 2002.

Leuchner, M., and Rappengluck, B.: VOC source–receptor relationships in Houston during TexAQS-II. Atmospheric Environment, 44, 4056–4067, 2010.

Lindinger, W., Hansel, A. and Jordan, A.: On-line monitoring of volatile organic compounds at pptv levels by means of Proton-Transfer-Reaction Mass Spectrometry (PTR-MS) Medical applications, food control and environmental research. International Journal of Mass Spectrometry and Ion Processes. 173, 191-241, 1998.

Link, M. F., B. Friedman, R. Fulgham, P. Brophy, A. Galang, S. H. Jathar, P. Veres, J. M. Roberts, and D. K. Farmer (2016), Photochemical processing of diesel fuel emissions as a large secondary source of isocyanic acid (HNCO), Geophys. Res. Lett., 43, 4033–4041, doi:10.1002/2016GL068207.

Roberts, J. M., Veres, P. R., Cochran, A. K., Warneke, C., Burling, I. R., Yokelson, R.

J., Lerner, B., Gilman, J. B., Kuster, W. C., Fall, R., and de Gouw, J.: Isocyanic acid in the atmosphere and its possible link to smoke-related health effects, Proceedings of the National Academy of Sciences, doi:10.1073/pnas.1103352108, 2011.

Sarkar, C., Sinha, V., Kumar, V., Rupakheti, M., Panday, A., Mahata, K. S., Rupakheti, D., Kathayat, B., and Lawrence, M. G., Overview of VOC emissions and chemistry from PTR-TOF-MS measurements during the SusKat-ABC campaign: high acetaldehyde, isoprene and isocyanic acid in wintertime air of the Kathmandu Valley, Atmos. Chem. Phys., 16, 3979-4003, 2016.

Sinha,V., Kumar,V., and Sarkar,C.: Chemical composition of pre-monsoon air in the Indo–Gangetic Plain measured using a new PTR-MS and air quality facility: high surface ozone and strong influence of biomass burning, Atmos. Chem. Phys., 14, 5921-5941, 2014.

Stockwell, C. E., Veres, P. R., Williams, J., and Yokelson, R. J.: Characterization of biomass burning emissions from cooking , peat, crop residue, and other fuels with high-resolution proton-transfer-reaction time-of-flight mass spectrometry, Atmos. Chem. Phys., 15, 845-865, doi:10.5194/acp-15-845-2015, 2015.

Taipale, R., Ruuskanen, T.M., Rinne, J., Kajos, M.K., Hakola, H., Pohja, T., and Kulmala, M.: Technical Note: Quantitative long-term measurements of VOC concentrations by PTR-MS – measurement, calibration, and volume mixing ratio calculation methods. Atmos. Chem. Phys., 8, 6681–6698, 2008.

Tolga, E., Cetin, B., Cetin, E., Bayram, A., and Odabasi, M.: Characterization of volatile organic compounds (VOCs) and their sources in the air of Izmir, Turkey. Environ. Monit. Assess. 133, 149–160, 2007.

Yokelson, R. J., Burling, I. R., Gilman, J. B., Warneke, C., Stockwell, C. E., de Gouw, J., Akagi, S. K., Urbanski, S. P., Veres, P., Roberts, J. M., Kuster, W. C., Reardon, J., Griffith, D. W. T., Johnson, T. J., Hosseini, S., Miller, J. W., Cocker Iii, D. R., Jung,

H., and Weise, D. R.: Coupling field and laboratory measurements to estimate the emission factors of identified and unidentified trace gases for prescribed fires, Atmos. Chem. Phys., 13, 89-116, doi:10.5194/acp-13-89-2013, 2013.

Yurdakul, S., Civan, M., and Tuncel, G.: Volatile organic compounds in suburban Ankara atmosphere, Turkey: Sources and variability. Atmospheric Research, 120, 298–311, 2013.

Wang, M., Shao, M., Chen, W., Yuan, B., Lu, S., Zhang, Q., Zeng, L., and Wang, Q.: A temporally and spatially resolved validation of emission inventories by measurements of ambient volatile organic compounds in Beijing, China. Atmos. Chem. Phys., 14, 5871–5891, 2014.

---

## Author Response (AR2)

**Responses to the Comments of the Editor**

**(1)** Page 4. The monitors that are used for $NO_x$ actually measure $NO_y$, because their catalytic convertors are not specific.
This is a good point. We added a sentence explaining the potential interferences to the measurements of such monitors using molybdenum oxide catalysts by other NOy components together with a reference to the work of Dunlea et al. (2007) for additional information.

Dunlea, E. J., Herndon, S. C., Nelson, D. D., Volkamer, R. M., San Martini, F., Sheehy, P. M., Zahniser, M. S., Shorter, J. H., Wormhoudt, J. C., Lamb, B. K., Allwine, E. J., Gaffney, J. S., Marley, N. A., Grutter, M., Marquez, C., Blanco, S., Cardenas, B., Retama, A., Ramos Villegas, C. R., Kolb, C. E., Molina, L. T., and Molina, M. J.: Evaluation of nitrogen dioxide chemiluminescence monitors in a polluted urban environment. Atmos. Chem. Phys., 7, 2691–2704, 2007.

**(2)** Page 6. The authors give numbers for sensitivities in ncps, that is not a sensitivity. Sensitivities are in units of ncps per concentration (or mixing ratio). In this case I assume those are ncps/ppbv? Please change that in the several places those appear.
These are indeed in ncps/ppb. We have made the corresponding correction.

**(3)** Page 6. I think you mean that your calibrator does not substantially change the RH of the air, not "does not make adjustments for the relative humidity".
We have rephrased this rather confusing statement following the editor's suggestion.

**(4)** Also page 6, you changed the sentence that describes formaldehyde and it looks like you inverted the meaning. I think you mean to say you did exclude formaldehyde, is that correct?
That is correct, the formaldehyde measurements were excluded. We have rewritten this sentence.